# Membrane cholesterol access into a G-protein-coupled receptor

Ramon Guixà-González[1,2,*], José L. Albasanz[3,*], Ismael Rodriguez-Espigares[1], Manuel Pastor[1], Ferran Sanz[1], Maria Martí-Solano[1], Moutusi Manna[4], Hector Martinez-Seara[4,5], Peter W. Hildebrand[2], Mairena Martín[3] & Jana Selent[1]

Cholesterol is a key component of cell membranes with a proven modulatory role on the function and ligand-binding properties of G-protein-coupled receptors (GPCRs). Crystal structures of prototypical GPCRs such as the adenosine $A_{2A}$ receptor ($A_{2A}R$) have confirmed that cholesterol finds stable binding sites at the receptor surface suggesting an allosteric role of this lipid. Here we combine experimental and computational approaches to show that cholesterol can spontaneously enter the $A_{2A}R$-binding pocket from the membrane milieu using the same portal gate previously suggested for opsin ligands. We confirm the presence of cholesterol inside the receptor by chemical modification of the $A_{2A}R$ interior in a biotinylation assay. Overall, we show that cholesterol's impact on $A_{2A}R$-binding affinity goes beyond pure allosteric modulation and unveils a new interaction mode between cholesterol and the $A_{2A}R$ that could potentially apply to other GPCRs.

[1] Research Programme on Biomedical Informatics (GRIB), Department of Experimental and Health Sciences of Pompeu Fabra University (UPF)—Hospital del Mar Medical Research Institute (IMIM), 08003 Barcelona, Spain. [2] Institut für Medizinische Physik und Biophysik, AG ProteiInformatics, Charité-Universitätsmedizin Berlin, Charitéplatz 1, D-10117 Berlin, Germany. [3] Department of Inorganic Chemistry, Organic Chemistry, and Biochemistry, Faculty of Science and Chemical Technologies and Faculty of Medicine of Ciudad Real. Regional Center of Biomedical Research (CRIB), University of Castilla-La Mancha (UCLM), 13071 Ciudad Real, Spain. [4] Department of Physics, Tampere University of Technology (TUT), PO Box 692, FI-33101 Tampere, Finland. [5] Institute of Organic Chemistry and Biochemistry, Academy of Sciences of the Czech Republic, CZ-16610 Prague, Czech Republic. * These authors contributed equally to this work. Correspondence and requests for materials should be addressed to M.M. (email: mairena.martin@uclm.es) or to J.S. (email: jana.selent@upf.edu).

G-protein-coupled receptors (GPCRs) are complex signalling machines that are embedded in the cell membrane. They are able to respond to extracellular signalling stimulus by triggering diverse intracellular pathways of high relevance for human biology. Despite tremendous advances in characterizing GPCR structure and activation mechanisms, relatively little is known about the role of the membrane environment or of specific membrane lipid composition in receptor function.

Recent work shows that membrane phospholipids can allosterically modulate the activity[1] and oligomerization[2] of GPCRs. In addition, membrane cholesterol significantly modulates the stability, ligand-binding properties and function of several GPCRs (reviewed in refs 3–6). Specifically, the presence of cholesterol in cell membranes can either enhance[7–11] (that is, positively modulate) or decrease[12–15] (that is, negative modulation) ligand binding and/or functional properties of different GPCRs (see Supplementary Table 1 for a comprehensive summary). A well-known example of this modulation is observed in rod outer segments for the prototypical receptor rhodopsin, where higher cholesterol concentrations in newly formed basal disks are used by these cells to stabilize the structure of metarhodopsin I (MI), thus hampering the formation of the active intermediate metarhodopsin II (MII)[16–19].

Whether this modulation is exerted through indirect effects[20,21] (that is, changes in membrane properties), direct interactions[22–25] between cholesterol and GPCRs, or both, has for long been a matter of intense debate (see ref. 6 for a recent review on this topic). Specific cholesterol-binding sites have been identified at the surface of different GPCRs[26], suggesting a potential allosteric role of cholesterol in modulating GPCR function. Intriguingly, other studies postulate that closely related cholesterol derivatives can even modulate the function of certain class-A GPCRs from the orthosteric binding pocket thus acting like conventional class-A GPCR ligands. For instance, oxysterol is thought to follow this binding mode at the Epstein–Barr virus-induced G-protein coupled receptor 2 (GPR183)[27] or the chemokine receptor CXCR2 (ref. 28). Similarly, oxysterol derivatives are known allosteric modulators of the oncoprotein Smoothened[29] (SMO), a class F GPCR. A very recent crystal structure of SMO shows one cholesterol molecule in the binding site of the extracellular domain of this receptor[30].

The adenosine $A_{2A}R$ receptor ($A_{2A}R$) is a class-A GPCR that plays a major role in the heart and brain by regulating oxygen consumption and blood flow[31]. In fact, in the central nervous system (CNS)[32], the $A_{2A}R$ constitutes a potential therapeutic target for the treatment of Alzheimer and Parkinson's disease[33,34]. Cholesterol binding to the $A_{2A}R$ at allosteric sites has been previously demonstrated by a high-resolution X-ray crystal structure (PDB ID 3EML)[24]. Computational work has further quantified allosteric cholesterol binding to the receptor surface[35,36] and suggested a stabilizing effect on the apo-form of the $A_{2A}R$[35]. However, the ability of cholesterol to impact ligand-binding properties at the $A_{2A}R$ remains unclear.

For this purpose, in the present study we analysed the influence of cholesterol depletion on ligand binding and studied the dynamics of cholesterol–$A_{2A}R$ interaction by extensive long-scale molecular dynamics (MD) simulations. Our simulation data reveals an unexpected mechanism of cholesterol action on ligand binding consisting on the entry of a cholesterol molecule into the receptor transmembrane bundle. Different lipophilic ligands that bind to the orthosteric site of class A GPCRs are suggested to access the protein from the membrane milieu (reviewed in ref. 6). Interestingly, in recent crystal

structures of rhodopsin[37,38], a molecule of a commonly used detergent (that is, $n$-octyl β-D-glucopyranoside) replaced retinal from the ligand-binding pocket. Therefore, as recently discussed by Gimpl[6], it would seem plausible that cholesterol can access the interior of class A GPCRs like the $A_{2A}R$. To validate this mechanism of action, we used a specifically tailored experimental approach to assess cholesterol impact on chemical modification of the $A_{2A}R$ interior. Taken together, our combined long-scale MD simulation and experimental results show that cholesterol can compete with orthosteric ligands by entering the receptor interior from the membrane side.

## Results

**Effect of cholesterol depletion on $A_{2A}R$ ligand binding.** To investigate the effect of membrane cholesterol on $A_{2A}R$-binding properties, we removed cholesterol from the membrane and monitored the specific binding of the radioligand [$^3$H]ZM241385 (Fig. 1), a selective antagonist of this receptor. We depleted membrane cholesterol by treating C6 glioma cells with methyl-β-cyclodextrin (MβCD), a specific cholesterol-sequestering agent, for time lengths between 0 and 50 min (Fig. 1). Our data indicate that MβCD is able to deplete around 70–80% of membrane cholesterol after 30 min. To accurately assess the level of cholesterol depletion, we carried out targeted lipidomics in plasma membranes. Remarkably, 40 min treatment with 5 mM MβCD according to the described protocol in the method section depletes up to 61% of cholesterol from the membrane (Supplementary Fig. 1). In addition, further radioligand binding assays using the former membrane preparations (Supplementary Fig. 2) confirm the effect we describe in Fig. 1 and Supplementary Fig. 3 using intact cells.

To rule out any cytotoxic effect of MβCD or WSC treatment, cell viability was determined at 20, 40 and 60 min after MβCD or WSC treatments using the XTT method (see Methods). As shown in Supplementary Fig. 4A, neither MβCD nor WSC treatments affected cell viability. Here it is worth to highlight that cells remain viable despite the depletion of more than 60% of their membrane cholesterol using the MβCD treatment

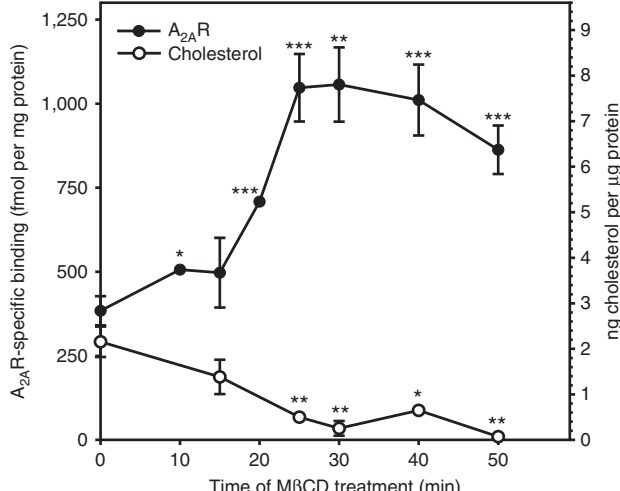

**Figure 1 | $A_{2A}R$-specific binding and cholesterol content.** Time course of 5 mM MβCD addition on [$^3$H]ZM241385-specific binding to $A_{2A}R$ in intact cells and membrane cholesterol content. This experiment was carried out using a saturating radioligand concentration of 40 nM. Mean ± s.e.m. values obtained from $n = 3$ separate experiments carried out in triplicate. *$P < 0.05$, **$P < 0.01$ and ***$P < 0.001$ significantly different from control value (time 0, $n = 5$) according to a Student's $t$-test.

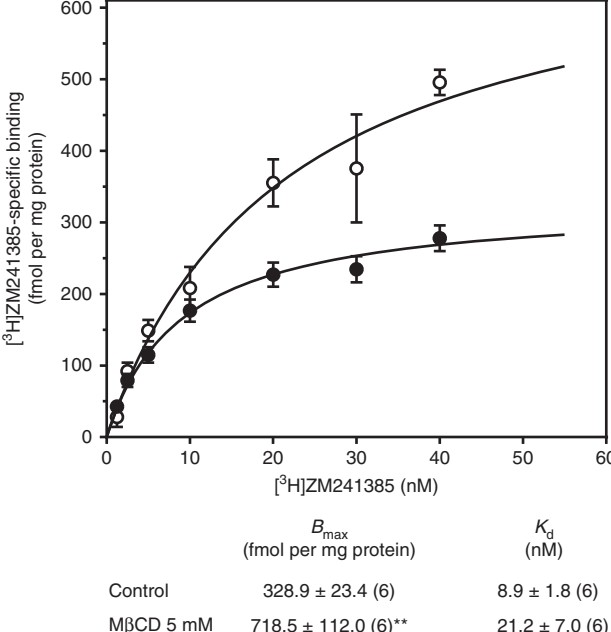

| | $B_{max}$ (fmol per mg protein) | $K_d$ (nM) |
|---|---|---|
| Control | 328.9 ± 23.4 (6) | 8.9 ± 1.8 (6) |
| MβCD 5 mM | 718.5 ± 112.0 (6)** | 21.2 ± 7.0 (6) |

**Figure 2 | Effect of MβCD on specific A₂AR binding in C6 intact cells.**
Control (closed circles) and 5 mM MβCD (40 min) (open circles) treated cells were incubated with different concentrations of [³H]ZM241385 as described in the Methods. These results are mean ± s.e.m. values obtained from six separate experiments carried out in duplicate. Kinetic parameters ($B_{max}$ and $K_d$) of the corresponding saturation binding curves are indicated at the bottom the figure. **$P < 0.01$ significantly different from control value according to a Student's $t$-test.

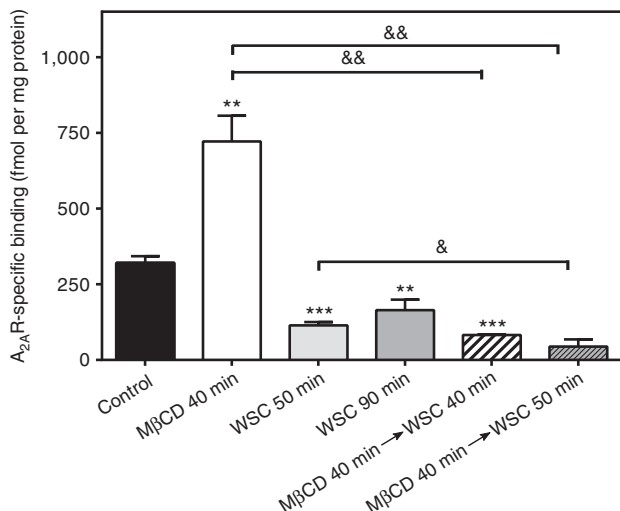

**Figure 3 | A₂AR-specific binding in intact cells.** A₂AR [³H]ZM241385 radioligand binding was determined after treatment with 5 mM methyl-β-cyclodextrin (MβCD), and/or 1 mM water soluble cholesterol (WSC) for 40, 50 or 90 min. Bars 5 and 6 represent a sequential treatment (that is, first MβCD is added, then a washing step, and finally WSC for 40 or 50 min). These experiments were carried out under saturating radioligand concentration (that is, 40 nM). Mean ± s.e.m. values obtained from $n = 3$ (columns 2, and 4–6), $n = 4$ (column 3), and $n = 5$ (column 1) separate experiments carried out in triplicate. *$P < 0.05$, **$P < 0.01$ and ***$P < 0.001$ significantly different from control value ${}^\&P < 0.05$ and ${}^{\&\&}P < 0.01$ significantly different from MβCD value according to a Student's $t$-test.

detailed in the Methods. Likewise, protein content did not significantly change after 20, 40 and 60 min incubation with either MβCD or WSC (Supplementary Fig. 4B). Moreover, cells did not display any significant change in number (Supplementary Fig. 5a), morphology (Supplementary Fig. 5b and Supplementary Movie 1), or division processes.

Remarkably, the lack of cholesterol increases the specific binding of [³H]ZM241385 to the A₂AR by more than 100% (30 min) when compared to non-treated cells (0 min) (Fig. 1 and Supplementary Fig. 3). Longer incubation times with MβCD (that is, 240 min) did not result in higher levels of A₂AR specific binding (469.7 ± 13.7 fmol per mg protein, $n = 2$), likely due to a compensatory mechanism to maintain cholesterol homeostasis in treated cells. Saturation binding experiments using a wide range of radioligand concentrations confirm this inhibitory effect (see Fig. 2). To validate the reversibility of this effect, we replenished membranes with cholesterol using water-soluble cholesterol (WSC) (Fig. 3).

Adequate cholesterol depletion and insertion into the membrane was monitored in intact cells and plasma membrane fractions by filipin fluorescence staining (Supplementary Figs 6 and 7) and targeted lipidomic analysis (Supplementary Fig. 1). Interestingly, addition of cholesterol significantly decreases [³H]ZM241385 binding in cell membranes either untreated (Fig. 3, columns 1, 3 and 4) or previously depleted from cholesterol using MβCD (Fig. 3, columns 2, 5 and 6). This clearly suggests that cholesterol has an inhibitory effect on [³H]ZM241385 binding to the A₂AR. This effect was confirmed in membranes from control cells by competitive binding experiments in the presence of increasing WSC concentrations (see Fig. 4 and Supplementary Note 1). To rule out that the former effect is the result of a higher number of A₂ARs

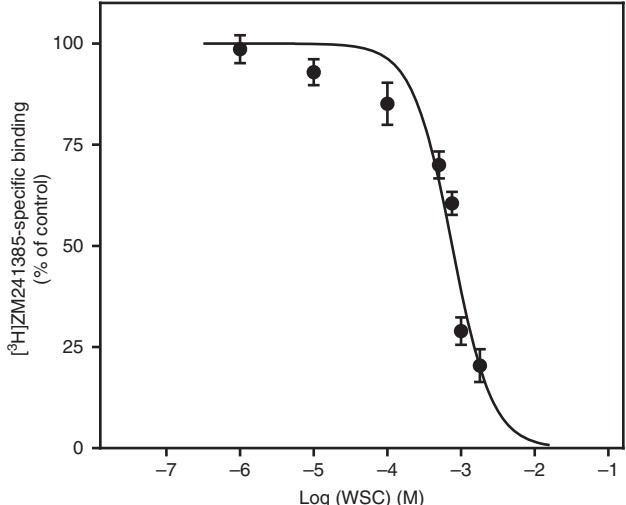

**Figure 4 | WSC competition binding curve in C6 plasma membranes.**
Plasma membranes isolated from control cells were incubated with 20 nM [³H]ZM241385 and different WSC concentrations (1 µM to 3 mM) as described in the Methods section. These results are mean ± s.e.m. values obtained from three different samples analysed in duplicate.

available due to an inhibition of receptor internalization by MβCD, we performed new binding assays in the presence of different inhibitors of endocytosis (see details in Supplementary Note 2). As shown in Supplementary Fig. 8, inhibiting endocytosis does not significantly modulate A₂AR specific binding, hence demonstrating that receptor internalization is not involved in the cholesterol-mediated modulation of A₂AR-specific binding. While cholesteryl hemisuccinate,

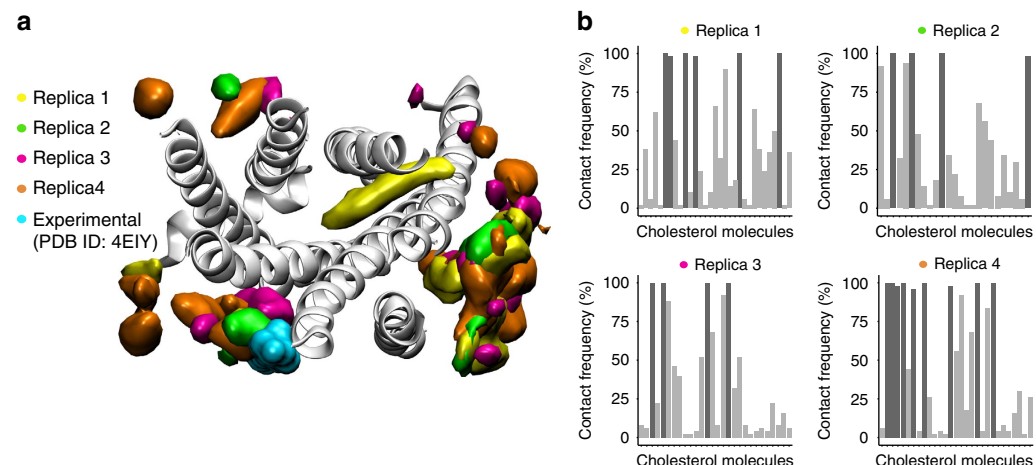

**Figure 5 | Cholesterol volumetric maps and contact frequency. (a)** Volumetric maps of cholesterol density around the aligned structure of the $A_{2A}R$ (white cartoon) for replica 1 (yellow), 2 (green), 3 (red) and 4 (orange). Density maps for individual replicas 1–4 can be also seen in Supplementary Fig. 10. Experimentally observed cholesterol molecule in the recently published high-resolution structure of Liu et al.[24] (PDB:4EIY) is shown in cyan surface. Protein is viewed from the extracellular side, helices are labelled and loops are not depicted for clarity. **(b)** Normalized contact frequency (%) (y axis) of cholesterol molecules (x axis) interacting with the $A_{2A}R$ (that is, below 2.9 Å) during each 1 μs trajectory (replicas 1, 2, 3 and 4). Here we consider cholesterol–$A_{2A}R$ binding interactions to be stable or transient when the normalized contact frequency is above (dark grey bars) or below (light grey bars) 95%, respectively.

a cholesterol derivative, enhances the stability and activity of detergent-solubilized $A_{2A}Rs^{39}$, our data clearly shows that naturally occurring cholesterol has an inhibitory effect on [³H]ZM241385 binding to the $A_{2A}R$ in more physiological environments (that is, intact cells and cell membrane preparations). Our findings go along with previous experimental evidence in several class A GPCRs reporting a negative cholesterol-mediated modulation of ligand binding (Supplementary Table 1).

**All-atom MD simulations of cholesterol interaction sites.** To shed light on the structural basis of the observed cholesterol modulation, we carried out a complete set of atomistic MD simulations of the $A_{2A}R$ (Supplementary Table 2). As described in the Methods section, we used a comprehensive set of lipids of different chains, nature and length to model a native-like lipid bilayer rich in cholesterol (Supplementary Table 3). The crystal structure of the $A_{2A}R^{40}$ was embedded in the former membrane and the system was solvated, neutralized, adjusted to an ionic strength of 150 mM $Na^+Cl^-$ and equilibrated following standard protocols (see Methods). First, we simulated a set of $4 \times 1\,\mu s$ replicas to study the frequency and stability of the interaction between cholesterol and the $A_{2A}R$. The analysis of the accumulated 4 μs shows that in average 12 cholesterol molecules are in contact (below 2.9 Å) with the $A_{2A}R$ throughout the simulation (Supplementary Fig. 9). This value is in agreement with the number of cholesterol molecules required for an ideal conformational stability of the $A_{2A}R$ experimentally observed in cholesterol-rich micelles[39].

While our simulations show several transient cholesterol–$A_{2A}R$ interactions (light grey bars, Fig. 5 right), certain cholesterol molecules establish permanent binding interactions to the receptor over nearly 100% of the simulation time (dark grey bars, Fig. 5 right). To spot preferred cholesterol interaction sites at the $A_{2A}R$ surface, we used a 3D volumetric map to depict the density of cholesterol molecules in the simulation (Fig. 5 left and Supplementary Fig. 10). In agreement with Lee et al.[36] our data show that transmembrane helices 2 and 3 (TM2–3),

TM3-4-5 and TM7-1 are preferred interaction areas of cholesterol at the $A_{2A}R$ surface. Notably, one of the predicted interaction sites, namely TM2–3, overlaps with one of the cholesterol binding sites shown in the experimental high-resolution structure of $A_{2A}R$ in complex with ZM241385 (ref. 24; Fig. 5 left). However, as described in ref. 36, cholesterol does not seem to significantly occupy the upper TM5–6 region, where another binding site was observed in the crystal structure.

**Cholesterol accesses the $A_{2A}R$ interior in MD simulations.** Strikingly, the volumetric analysis also shows high cholesterol density inside the transmembrane bundle of the $A_{2A}R$ (Fig. 5, left) indicating that cholesterol entered the $A_{2A}R$ from the membrane milieu. A visual inspection of the individual trajectories shows that one cholesterol molecule spontaneously accesses the interior of the protein from the extracellular leaflet through helices TM5–6 occupying a key area of the orthosteric binding pocket (see replica 1 in Fig. 5 and Supplementary Movies 2 and 4). Although cholesterol has been shown to occupy deeply buried sites in other membrane proteins[41], this is the first dynamic view of membrane cholesterol spontaneously invading the orthosteric binding pocket of a GPCR.

To exclude simulation artefacts and better explore the cholesterol entry pathway, we performed new simulations and studied the tendency of cholesterol to access the $A_{2A}R$. First, we selected four representative snapshots from the original cholesterol entrance trajectory (that is, replica 1) prior to the complete invasion of the receptor (Fig. 6). Then, each starting point was used to re-spawn 10 new trajectories of 100 ns each (that is, $10 \times 4 \times 100$). To quantify cholesterol progression towards the interior of the protein, we monitored the distance between cholesterol and residue E1.39 (Fig. 6). In most of these short trajectories, cholesterol does not back away from the receptor but it stays bound or progresses towards the interior of the receptor. As shown in Fig. 6 and Supplementary Table 4, cholesterol progression is much faster once cholesterol slightly tilts down (Fig. 6a,b) adopting a favored position to enter the receptor. Inclusion of intracellular loop 3 (ICL3) do not

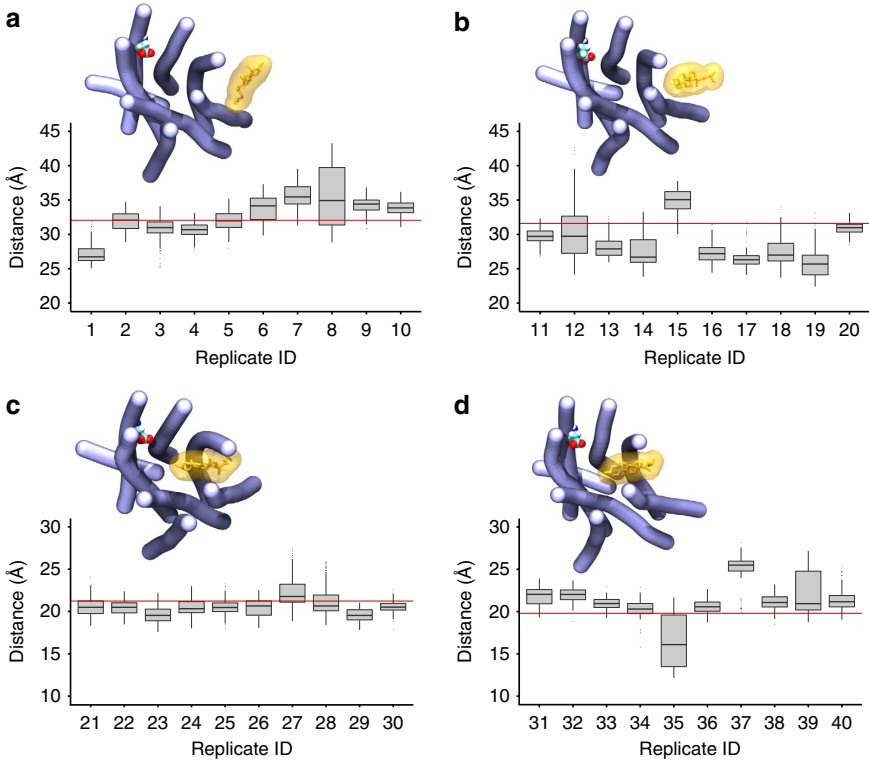

**Figure 6 | Short simulation replicas of cholesterol entrance.** Boxplots display the distance between the centre of mass of cholesterol and residue E1.39 for a set of 40 replicate simulations of 100 ns. Four different starting positions (**a–d**) re-spawned from the original cholesterol entrance trajectory were used to run each 10 replicas (that is, 1–10, 11–20, 21–30, 31–40). The distance at the beginning of the simulation as measured from the snapshot used to re-spawn each set of 10 trajectories (that is, **a–d**) is represented as one single red horizontal line in each of the graphs. Average distance for each set of replicates is reported in Supplementary Table 4. Inset figures show the initial structure of the A$_{2A}$R (in blue) and cholesterol residue (in orange) used to start each set of simulations. E1.39 residue is displayed as van der Waals spheres. The BENDIX[75] plugin for VMD was used to depict protein helices. Protein loops were omitted for clarity.

have a significant effect (Supplementary Fig. 11). These data clearly suggest that cholesterol can spontaneously access the A$_{2A}$R through a portal gate between TM5 and 6.

The nature of the membrane environment could be one of the driving forces behind the spontaneous cholesterol entrance into the A$_{2A}$R. As shown in Supplementary Movie 3, an unsaturated phospholipid (in yellow) along with a cluster of four cholesterol molecules seem to influence cholesterol entrance by preventing it from diffusing back to the membrane bulk. To study the impact of membrane composition, we substituted the compact membrane environment used so far by a pure 1-palmitoyl-2-oleyl-sn-glycero-3-phosphocholine (POPC) bilayer (see Methods) leaving intact both the target cholesterol and the A$_{2A}$R (see Methods). Interestingly, in the absence of a more compact and thicker membrane (Supplementary Fig. 12), cholesterol progression towards the interior of the protein is significantly diminished after 100 ns (see Supplementary Fig. 13 and Supplementary Table 4). Therefore, our simulations suggest that the ability of cholesterol to access the interior of the A$_{2A}$R can be modulated by the nature of the membrane environment.

The sequence of events during cholesterol entrance is depicted in Fig. 7. Initially, cholesterol is interacting with other membrane lipids (not shown) and water molecules (Fig. 7a) until it leaves this preferred position plunging its polar head into the hydrophobic core of the membrane (Fig. 7a,b). Next, cholesterol descends along TM5–6 guided by the formation of a hydrogen bond with the hydroxyl group of Y5.411 side chain (Fig. 7b). At this point, TM5 and TM6 are tightly packed involving an

aromatic cluster of staggered residues F5.45, F4.44, H6.52 and W6.48 (red arrows, Fig. 7b). Thereafter, cholesterol tilts 90° and pushes aside the aromatic side chains of residues F5.45 and F5.44 (Fig. 7c,d) attracted by water molecules and residue E169 at the extracellular loop 2 (ECL2) (Fig. 7c). This creates a protein gateway between TM5 and TM6 that cholesterol uses to make its way into the A$_{2A}$R to interact with E169 both directly and indirectly through contact with water molecules (dashed red lines in Fig. 7c). Finally, cholesterol completely enters the receptor attracted by E1.39 at TM1 (Fig. 7d). At this stage, the polar head engages into a hydrogen bonding network formed by Y7.36, E1.39 and water molecules. During the entrance of cholesterol, the communication between protein residues in the aromatic network is partially disrupted (F5.45–F4.44 and F4.44–H6.52, see red crosses in Fig. 7c,d), thus likely hampering the reverse progression of cholesterol towards the membrane bulk. However, cholesterol intercalation between helices TM5 and 6 does not involve marked protein rearrangements (Supplementary Movies 2–4).

To better characterize the behaviour of cholesterol inside the receptor, we extended three of the original 100 ns replicas (namely replicas 1, 35 and 38 in Fig. 6) up to 10 μs (3 × 10 μs). Once inside the receptor, cholesterol molecules explore the interior of the protein by establishing transient interactions rather than adopting a stable binding pose. Interestingly, our simulations show that cholesterol highly populates a specific area of the A$_{2A}$R binding pocket (yellow surface, Fig. 8a and Supplementary Movie 4) that highly overlaps with the classical orthosteric binding site and the position of the ZM241385 ligand

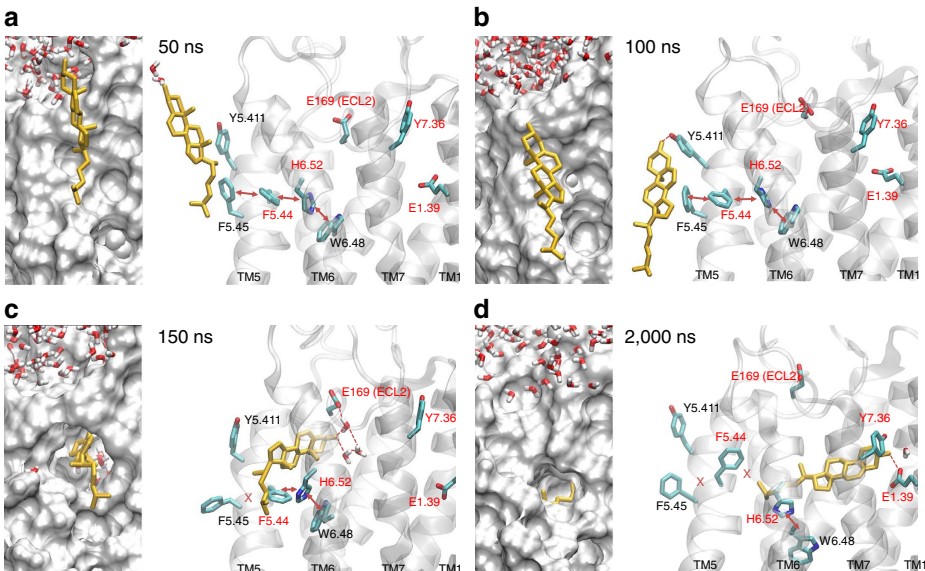

**Figure 7 | Cholesterol entrance. (a–d)** Four snapshots from a 2 μs MD trajectory showing cholesterol entrance through helices TM5–6 of the A$_{2A}$R. Left panel: view from the membrane side towards the receptor surface (grey surface), right panel: detailed structural representation of relevant residues that interact with cholesterol during its penetration into the receptor.

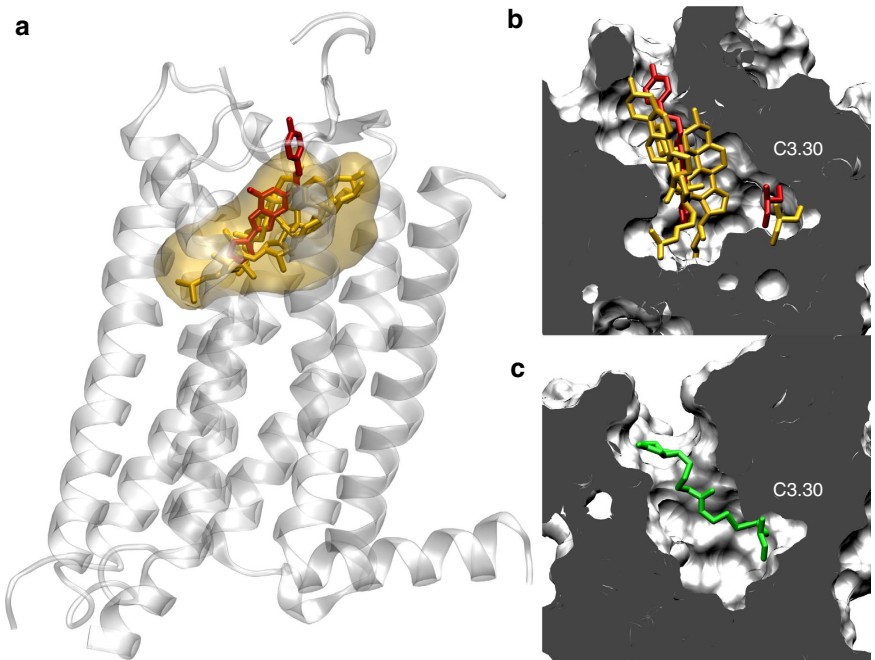

**Figure 8 | Cholesterol behaviour inside A$_{2A}$R in long-scale MD simulations. (a)** Average position of cholesterol in the orthosteric binding site calculated over the accumulated 3 × 10 μs (yellow transparent map) superimposed onto the crystallized A$_{2A}$R in complex with the ZM241385 antagonist (red sticks, PDB:3EML). A single snapshot of cholesterol position at the end of each 10 μs simulation is depicted in yellow sticks. **(b)** Position of C3.30 in the binding site crevice with respect to ZM241385 and cholesterol molecules (position at 10 μs of three individual MD trajectories). **(c)** Model of the C3.30 chemically modified with MTSEA-B in the A$_{2A}$R binding site crevice.

bound to the A$_{2A}$R crystal structure (PDB:3EML) (red sticks, Fig. 8a). Consequently, our data indicate that, if cholesterol reaches the interior of the protein, it will likely hamper the binding of A$_{2A}$R ligands such as ZM241385 to the orthosteric binding site. This finding represents one plausible mechanism behind the observed increase in [$^{3}$H]ZM241385 binding upon cholesterol depletion.

**Experimental validation of cholesterol occupying A$_{2A}$R.** Providing experimental evidence for a native membrane lipid occupying the interior of a GPCR is a challenging mission. Cholesterol has been shown to bind the A$_{2A}$R surface in recent X-ray data[24]. However, there is currently no structural hint or experimental evidence pointing towards cholesterol ability to occupy the interior of this receptor. In this study, in an

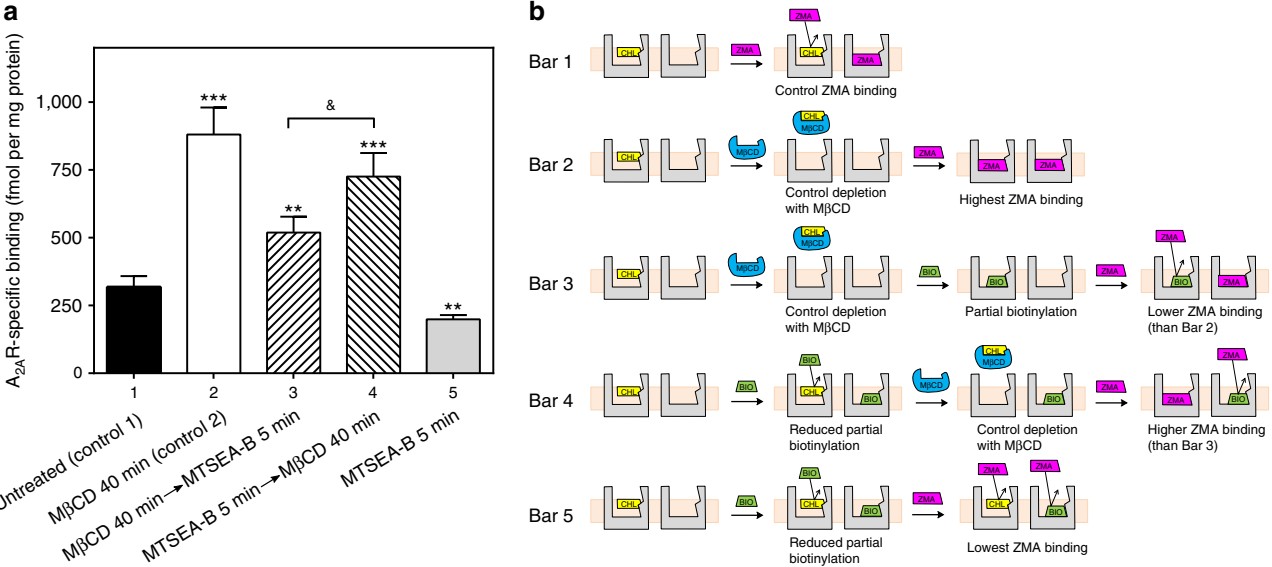

**Figure 9 | Biotinylation experiments. (a)** Effect of the MTSEA-B reagent and cholesterol depletion on [$^3$H]ZM241385-specific binding (mean ± s.e.m.) at a saturating radioligand concentration of 40 nM. **(b)** Scheme of cholesterol influence on receptor biotinylation according to the experimental results presented in **a**. Bar 1 (control, $n = 11$): ZMA binding in untreated (cholesterol-containing) control cells yields low ZMA binding due to cholesterol shielding effect. Bar 2 ($n = 5$): ZMA binding in MβCD-treated cell membranes (cholesterol-depleted) results in highest ZMA binding due to an empty binding pocket. Bar 3 ($n = 4$): biotinylation of a cholesterol-depleted binding pocket results in a high degree of biotinylation in the orthosteric binding pocket. As the binding pocket is occupied by the biotinylated side chain of C3.30, ZMA binding is reduced. Bar 4 ($n = 4$): low biotinylation degree due to the shielding effect of cholesterol in the binding pocket. Subsequent removal of the shielding agent cholesterol with MβCD results in an empty binding pocket that allows for high ZMA binding. Bar 5 ($n = 4$): lowest ZMA binding due to the accumulated hindering effect of biotinylation and cholesterol. CHL, cholesterol (yellow); BIO, biotinylation (green); MβCD, methyl-β-cyclodextrin (blue); ZMA, ZM241385 ligand (red). In **a** **$P < 0.01$ and ***$P < 0.001$ significantly different from control 1, $^\&P < 0.05$ significantly different between them (bars 3 and 4) according to a Student's $t$-test.

attempt to provide experimental evidence supporting our computational findings, we adapted a biochemical assay originally used in GPCRs by Javitch *et al.*[42] for scanning residues exposed to the binding-site crevice in the dopamine D$_2$ receptor. This assay takes advantage of the fact that thiol groups of cysteine residues facing the binding-site crevice of a GPCR will selectively react with hydrophilic and sulfhydryl-specific reagents such as derivatives of methanethiosulfonate (MTS) (detailed in Supplementary Note 3). As shown by the A$_{2A}$R crystal structure (PDB: 3EML)[24] (see Supplementary Fig. 14 and Supplementary Table 5), water-accessible cysteine residues with a free sulfhydryl group are only found in the interior of the receptor. In our experiments, we chemically modified by biotinylation these water-exposed cysteines using *N*-biotinylaminoethyl–methanethiosulfonate (MTSEA-B), a positively charged MTS derivative similar in size and molecular weight to the ZM241385 ligand (see Methods). Figure 8c illustrates that biotinylation of residue C3.30 (green sticks) results in a modified side chain that occupies a large region of the orthosteric binding site that overlaps with the preferred binding site of cholesterol as well as of ZM241385 (yellow and red sticks, respectively, Fig. 8b). As shown in Supplementary Fig. 15, chemically modified cysteine residues C5.46 and C6.56 yield a similar overlap with the ZM241385 ligand. Accordingly, if cholesterol is able to invade the A$_{2A}$R-binding pocket as proposed by our simulations, cysteine biotinylation should be hindered in cholesterol-rich membranes compared to cholesterol-depleted membranes.

Based on this conceptual framework, we designed a new set of radioligand binding experiments using MTSEA-B to covalently modify cysteine residues in the A$_{2A}$R interior and MβCD to deplete membrane cholesterol (Fig. 9). Experiments are

schematically depicted in Fig. 9b. First, we used two control experiments where we measured [$^3$H]ZM241385 in untreated (Bar 1) and cholesterol-depleted cells (Bar 2). These control experiments corroborated that cholesterol depletion by MβCD favours specific binding of [$^3$H]ZM241385 (Bar 2) by three fold when compared with untreated conditions (Bar 1). Then, we assessed if cysteine residues in the A$_{2A}$R interior are susceptible to MTSEA-B biotinylation in cholesterol-depleted conditions, where no competition between cholesterol and [$^3$H]ZM241385 is expected. We observe that after cholesterol depletion, biotinylation reduces [$^3$H]ZM241385 binding by ~40% (Bar 3) when compared with treatment with MβCD alone (Bar 2).

This marked reduction in specific binding strongly indicates that at least one of the cysteine residue in the A$_{2A}$R interior (Supplementary Note 3) is susceptible to biotinylation and confirms that its chemically modified side chain occupies the orthosteric-binding site (Fig. 9c). In the next measurement, we tested the influence of cholesterol binding in cysteine biotinylation (Bar 4). Remarkably, [$^3$H]ZM241385 binding increases by about 25% (Fig. 9, Bars 3–4) when receptors are treated with MTSEA-B prior to cholesterol depletion. This suggests that cholesterol has a shielding effect by protecting cysteine residues from biotinylation inside the receptor. Thus, this observation supports the presence of a cholesterol molecule inside the A$_{2A}$R transmembrane bundle with the ability to block access of MTSEA-B to water-accessible cysteine residues. Finally, we observe that if the system is biotinylated in the presence of cholesterol and cholesterol is not removed afterwards (Bar 5), there is an additive effect of cholesterol and biotinylation that hampers binding of [$^3$H]ZM241385, yielding less specific binding than in control conditions (Bar 1).

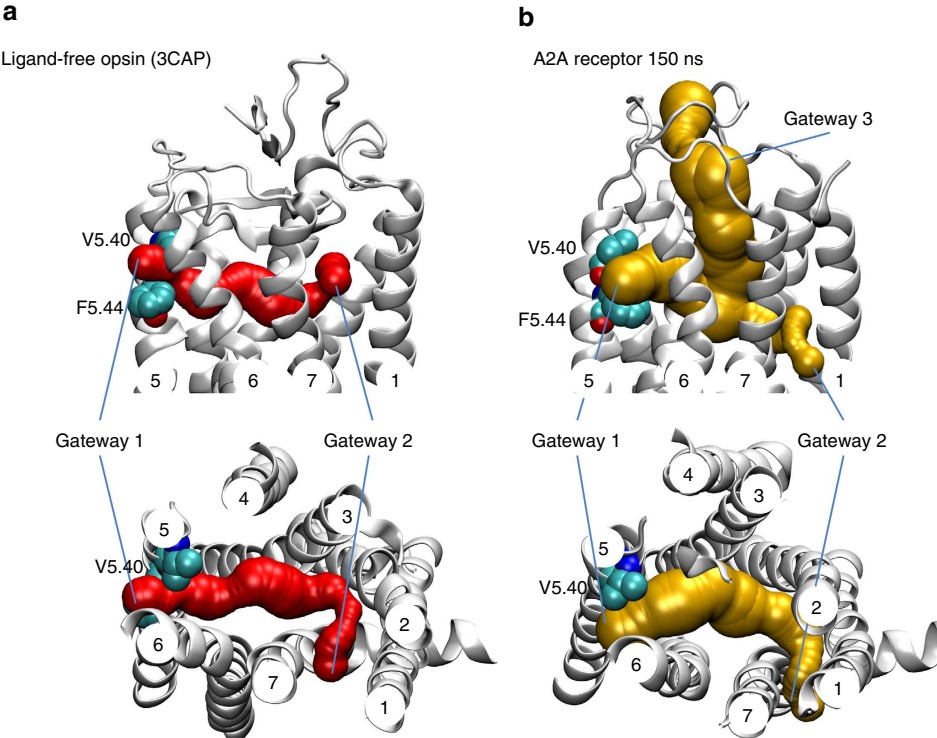

**Figure 10 | Tunnel pathway comparison between opsin and A$_{2A}$R.** Gateways and tunnel pathways for (**a**) the high-resolution structure opsin (3CAP)—red surface and (**b**) the simulated A$_{2A}$R (150 ns)—yellow surface computed using the Caver software[49]. Top: view from the membrane side, bottom: view from the extracellular side showing omitting gateway 3 for clarity in the A$_{2A}$R.

**Cholesterol entry/exit pathways**. Our data strongly suggest that a cholesterol molecule can penetrate the A$_{2A}$R from the membrane side. In fact, membrane access of molecules to class-A GPCRs has been reported for few other cases such as the lipid receptor S1P1 (ref. 43), the cannabinoid CB2 receptor[44] and the opsin receptor[45]. Among them, retinal uptake into the opsin receptor is probably the best-studied case where two potential gateways for ligand uptake or release have been described[45]. The larger opening (gateway 1) between helices 5 and 6 has initially been proposed to allow for retinal loading, whereas a smaller gate between helices 1 and 7 (gateway 2) allows for ligand exit[46]. In contrast, more recent studies point to a retinal entrance/exit in the opposite sense[47,48]. The complete tunnel pathway for retinal into the binding pocket of opsin including gateways 1 and 2 can be detected using the Caver software[49] (red surface, Fig. 10a). In comparison, Caver computation for the simulated A$_{2A}$R (150 ns) reveals three gateways 1 to 3 (yellow surface, Fig. 10b). The last largest gateway 3 is the classical channel for ligand entrance from the extracellular side into the receptor. The smallest gateway 2 is directed towards helices 1 and 7 whereas the intermediate gateway 1 is located between helices 5 and 6 (yellow surface, Fig. 10b). Remarkably, we find that gateway 1 is flanked by two residues, namely V5.40 and F5.44, which are conserved between the A$_{2A}$R and opsin (Fig. 10a,b). Hence, our data indicate that the opening that cholesterol uses to penetrate into the A$_{2A}$R is identical to the opening that retinal takes to enter/exit the opsin receptor.

Unveiling the putative exit pathway of cholesterol from the A$_{2A}$R is currently beyond the reach of all-atom unbiased simulations. If cholesterol behaved in a similar way to classical GPCR ligands, this entry/exit pathway could occur via the extracellular domain, as suggested elsewhere[28,50]. However, due to the large hydrophobic moiety of the cholesterol

molecule, it is not likely that cholesterol abandoning the A$_{2A}$R-binding pocket via the extracellular aqueous phase is energetically favoured. To determine the energetic cost of such exit route, we carried out a set of biased simulations (see Methods) where we computed the free energy profile (PMF) of extracting cholesterol from the binding site into the extracellular water phase. These simulations confirm that the energetic cost of a complete cholesterol exit from its observed binding site to the water phase is as high as 120 kJ·mol$^{-1}$ (Supplementary Fig. 16). Hence, it is more likely that cholesterol leaves the A$_{2A}$R via the transmembrane helices. In this context, we find that TM2-TM1 or TM7-TM1 could be possible cholesterol exit ways, as detailed in Supplementary Note 4. Notably, TM7-TM1 exit is in line with A$_{2A}$R channels that have been computed using the Caver software (Fig. 10b) as well as with the described gateway 2 in the crystallized structure of opsin (Fig. 10a)[47,48].

**Discussion**

In the present work, we have studied the effect of membrane cholesterol on the ligand-binding properties of the A$_{2A}$R. Our *in vitro* experiments demonstrate that cholesterol significantly decreases the binding of the antagonist [$^3$H]ZM241385 to the A$_{2A}$R. Molecular insights obtained in this and other studies point to numerous allosteric binding sites at the A$_{2A}$R surface that could potentially be involved in reducing ZM241385 binding. However, our work reveals an additional and unexpected mode of cholesterol action. We have observed that cholesterol modulates orthosteric ligand binding at the A$_{2A}$R after entering from the membrane side. These results have been validated combining both computational and specifically tailored experimental approaches.

Our study suggests that the gateway that cholesterol uses to enter the $A_{2A}R$ is evolutionarily conserved and identical to the retinal gateway present in the opsin receptor. Since molecules of a similar size and amphipathic nature have been shown to access the retinal-binding pocket[37], it is reasonable to speculate that the negative modulation exerted by cholesterol on rhodopsin function[16–19] could be partly mediated by the new mechanism proposed here. Taken all together, cholesterol is likely to modulate $A_{2A}R$ ligand-binding properties through a mixed mode of action, namely both orthosteric- and allosterically. Nevertheless, further research shall be aimed to unveil the exact contribution of each mode of cholesterol-mediated modulation at this or other GPCRs.

All in all, our results provide an important advance in the understanding of the interplay between cholesterol and the $A_{2A}R$. Ultimately, our work highlights the importance of considering the influence of different levels of membrane cholesterol in GPCR function and stresses the importance of accounting for these particular cholesterol effects in the study of cardiovascular and CNS disorders. From a broader perspective, our work opens the door to new studies on the effects of cholesterol and other endogenous lipids in GPCRs modulation. This knowledge paves new roads for exploring potential therapeutic uses of membrane sterols or sterol-mimetic molecules in GPCR drug discovery.

## Methods

**Cholesterol and MTSEA-B experiments.** Materials—[³H]ZM241385 ([2-³H](4-(2-[7-amino-2-(2-fury1) [1,2,4] triazolo [2,3-α] [1,3,5] triazin-5-ylamino]ethyl)phenol 27.4 Ci mmol⁻¹) was purchased from ARC (St Louis, MO, USA). The reagents MβCD, WSC, MTSEA-B and Theophylline were acquired from Sigma Aldrich (Madrid, Spain). Dynasore (3-Hydroxynaphthalene-2-carboxylic acid (3,4-dihydroxybenzylidene)hydrazide) and Pitstop 2 (N-[5-(4-bromobenzylidene)-4-oxo-4,5-dihydro-1,3-thiazol-2-yl]naphthalene-1-sulfonamide) were from Abcam Biochemicals (Cambridge, UK). Liquid scintillation cocktails were supplied by Perkin Elmer (Boston, MA, USA). Additional reagents were of analytical grade.

**Cell culture.** C6 cells from rat glioma were obtained from the American Type Culture Collection (ATCC). Dulbecco's modified Eagle's medium was supplemented with 2 mM L-glutamine, 10% fetal calf serum, 1% nonessential amino acids and antibiotics. Cells were maintained at 37 °C in a humidified atmosphere of 95% air and 5% CO₂ (ref. 51).

**Plasma membrane isolation.** Cells were homogenized on ice-cold isolation buffer (50 mM Tris-HCl pH 7.4, 10 mM MgCl₂-containing protease inhibitors) and centrifuged at 4 °C for 5 min at 1,000g in a Beckman JA 21 centrifuge. The supernatant was centrifuged at 4 °C for 20 min at 27,000g and the pellet was resuspended in isolation buffer[51].

**Radioligand binding assays.** Specific binding to $A_{2A}R$ was assayed in intact C6 cells in the different conditions analysed. Intact cells grown in 24-well plates were incubated with 40 nM [³H]ZM241385 (that is, saturating ligand concentration, based in our previous work[52]). Adenosine deaminase (ADA, 5 U ml⁻¹) was used to remove endogenous adenosine. [³H]ZM241385 ranging from 1.25 to 40 nM was utilized in saturation binding assays, where 5 mM theophylline was used to obtain non-specific binding. After 2 h at 25 °C, cells were washed with ice-cold culture medium, lysed with 0.2% SDS, and transferred to vials to count radioactivity. Two wells from each plate were employed for protein content measurement. When binding assays were performed in plasma membranes, fifty to one hundred micrograms of protein were pre-incubated with 5 U ml⁻¹ ADA for 30 min at 25 °C and maintained for 2 h at 25 °C in the presence of 20 or 40 nM [³H]ZM241385. Non-specific binding was also obtained with 5 mM theophylline, as in previous work[52]. For competition binding experiments, different (1 μM–3 mM) concentrations of WSC were used to displace total binding of 20 nM [³H]ZM241385. Binding to plasma membranes was finished by rapid filtration through Whatman GF/B filters, which were immediately washed and counted. Radioactivity measurements in vials or filters were performed in a Microbeta Trilux liquid scintillation counter (Wallac).

**Cholesterol analysis in intact cells.** Cholesterol content in intact C6 cells was measured with a Cholesterol Quantitation kit (MAK043) from Sigma

(Madrid, Spain), following manufacturer's instructions. Briefly, samples (10⁶ cells) were extracted with 200 μl of chloroform:isopropanol:IGEPAL CA-630 (7:11:0.1) in a microhomogenizer. After centrifugation at 13,000g for 10 min to remove insoluble material, the organic phase of samples (160 μl) was transferred to a new tube and dried at 50 °C in a SpeedVac for 30 min to remove chloroform. Dried lipids were then dissolved with 200 μl of the Cholesterol Assay Buffer. Fifty microlitres of samples and standards (1–5 ng) were added to 50 μl of reaction mixture and absorbance at 570 nm measured after 60 min incubation at 37 °C.

**Filipin fluorescence staining.** A cell-based Cholesterol Assay Kit from Abcam (Cambridge, UK) was performed in intact C6 cells in order to visualize and measure cholesterol by using Filipin III as a fluorescence probe of cholesterol. Briefly, after removal of culture medium from wells, cells were fixed for 10 min and washed (3 × 5 min). Filipin III solution was added to each well assayed and maintained in the dark for 45 min at room temperature. After washing of cells (2 × 5 min) fluorescence images were obtained with a digital camera (Leica DFC350FX), attached to a Leica DMI6000B (Leica Microsystems, Wetzlar, Germany) fluorescent microscope using × 20 HCX PL FLUOTAR objective.

**Depletion and reloading of cholesterol in living C6 cells.** To extract cholesterol, the cells were incubated with 5 mM MβCD for indicated period of time at 37 °C. Cholesterol enrichment of the cells was started using 1 mM WSC for indicated period of time at 37 °C. Finally, chemical modification of the $A_{2A}R$-binding pocket by biotinylation was performed with N-biotinylaminoethylmethanethiosulfonate (MTSEA-B). This compound was dissolved in DMSO, and aliquots of 100 mM stock solution were thawed just prior to use. When needed, cells were incubated in 500 μM MTSEA-B for 5 min at 37 °C. In all cases, treatment was performed in serum-free medium.

**Lipidomic analysis.** Plasma-membrane preparations were isolated from control (n = 3), 5 mM MβCD 40 min (n = 2) and 1 mM WSC 50 min (n = 2) treated cells. Membrane samples containing deuterated cholesterol D7 as internal standard were mixed with an equal volume (0.1 ml) of methanol and two volumes (0.2 ml) of chloroform. After each addition tubes were vortexed for 10 s. Chloroform phase (lower) was transferred to a glass tube after centrifugation for 15 min at 4 °C and at 1,000g. This last step was repeated twice. The chloroform phase was evaporated in a Speed Vac (Thermo Fisher Scientific, Barcelona, Spain) and resuspended in 50 μl of methanol:chloroform (3:1)[53,54]. These lipid extracts (2 μl) were analysed by mean of mass-spectrometry using a HPLC 1290 series coupled to an ESI-Q-TOF MS/MS 6520 (Agilent Technologies, Barcelona, Spain). LC/MS analysis required an XBridge BEH C18 shield column (100 mm × 2.1 mm ID × 1.7 μm) from Waters (Milford, MA, USA) kept at 80 °C. Mobile phases (0.5 ml min⁻¹) consisted of 20 mM ammonium formate (pH 5) (A) and methanol (B). The gradient profile was: 50–70% B in 14 min, 70–90% B in 50 min, isocratic separation of 90% B during 15 min, 90–100% B in 5 min, and maintained so for an additional 5 min[55]. This protocol allowed the orthogonal characterization of lipids based on exact mass (<10 p.p.m.) and on retention time features. Collection of data was achieved in both positive and negative electrospray ionization time-of-flight modes and performed in full-scan mode at 100–3,000 m/z in an extended dynamic range (2 GHz), using N₂ (5 l min⁻¹, 300 °C) as nebulizer gas. The capillary voltage was 3,500 V (1 scan per s). Data were recorded and analysed by MassHunter Data/Qualitative analysis software (Agilent Technologies, Barcelona, Spain) to obtain the molecular features of the samples[56].

**Inhibition of endocytosis in living C6 cells.** To inhibit endocytosis, C6 cells were incubated with 80 μM Dynasore, a cell-permeable dynamin inhibitor, or 25 μM Pitstop 2, for 20 or 40 min at 37 °C (see Supplementary Note 3 for more details).

**Cell viability assay.** Cells were seeded (10⁴ cells per well) and grown in 96-well tissue culture plate and incubated with 0.3 mg ml⁻¹ XTT solution (sodium 3′-[1-(phenylaminocarbonyl)-3,4-tetrazolium]-bis (4-methoxy-6-nitro) benzene sulfonic acid hydrate) for 30 min at 37 °C in control, MβCD- or WSC-treated cells. The cleavage of XTT to form an orange formazan dye by viable cells was monitored by reading absorbance at 475 and 690 nm according to the manufacturer's protocol (Cell Proliferation Kit II, Roche, Mannheim, Germany).

**Protein determination.** Protein concentration was measured by the Lowry method, using bovine serum albumin as standard.

**Statistical analysis.** The binding data were analysed using Student's t-test, one-way analysis of variance and nonlinear regression fitting to saturation:

$$Y = B_{max}X(K_d + X)^{-1} \qquad (1)$$

or competition:

$$Y = 100\left(1 + 10^{((\text{LogEC50} - X)\text{HillSlope})}\right) \qquad (2)$$

binding curves with the GraphPad Prism 5.0 program (GraphPad Software, San Diego, CA, USA). Differences between mean values were considered statistically significant at $P < 0.05$.

**Water-accessible cysteine residues in the A$_{2A}$R.** According to high-resolution crystallography data[40], water-accessible cysteine residues with free sulfhydryl groups are only found in the interior of the receptor as all extracellular water-accessible cysteines are engaged in disulfide bonds and therefore cannot react with the biotinylation reagent (see detailed description in Supplementary Note 3). Among these residues, C3.30 is the closest one to the ZMA ligand and is therefore used as representative in this manuscript (Fig. 8c).

**Classical (non-biased) MD simulations.** All non-biased simulations were performed using the ACEMD simulation package[57]. The CHARMM36 (ref. 58), CHARMM36c (ref. 59) and CHARMM27 (ref. 60) force fields were used to represent lipids, cholesterol and proteins, respectively. A list of all simulations performed in this work is displayed in Supplementary Table 2.

*Set-up.* First, a multicomponent lipid bilayer of $\sim 100 \times 100\,\text{Å}^2$ (in the membrane plane) was built using the CHARMM-GUI membrane builder[61]. The exact lipid composition is described below. This membrane was equilibrated for 1 μs in the NPT ensemble. Next, the crystal structure of the adenosine A$_{2A}$R (PDB:3EML)[40] was manually embedded into the equilibrated membrane using VMD1.9 (ref. 62). The intracellular loop 3 (ICL3) of the A$_{2A}$R, not resolved in any of the current GPCR crystal structures, was omitted and all titratable residues were left in the dominant protonation state at pH 7. Next, an NPT equilibration phase was performed to accommodate water and lipid molecules to the protein and to promote further mixing (see below).

*Initial production runs.* Each of the equilibrated replicas above was simulated for 1 μs in the NVT ensemble (simulation set 1 in Supplementary Table 2).

*Short replicas.* Four different snapshots of the specific trajectory where cholesterol enters the receptor (see Results section) were manually selected (Fig. 6). Each of these snapshots served as a starting seed for 10 new replicas ($4 \times 10$) that were run for 100 ns in the NVT ensemble (simulation set 2 in Supplementary Table 2). To rule out an effect of ICL3 omission, we performed a similar set of simulations where the ICL3 was included (see simulation protocol below and Supplementary Fig. 11). To validate the membrane effect on cholesterol entrance, we substituted all membrane lipids by POPC. First, four new structures were generated by removing all lipid molecules from the starting four seeds used above. The CHARMM-GUI membrane builder[61] was then used to embed each structure into a pure POPC bilayer of $\sim 100 \times 100\,\text{Å}^2$. Systems were then minimized and equilibrated (see below). Similarly, each seed was used to simulate 100 ns of $4 \times 10$ replicas in the NVT ensemble (simulation set 3 in Supplementary Table 2). To rule out an effect of ICL3 omission, we performed a similar set of simulations where the ICL3 was included (see below).

*Long replicas.* To study the behaviour of cholesterol inside the receptor, we chose 3 of the 100 ns trajectories where cholesterol seemed to have an effective progression towards the interior of the receptor (that is, replicas 1, 35 and 38 in Fig. 6). These simulations were run for 10 μs in the NVT ensemble (simulation set 5 in Supplementary Table 2).

*System composition.* We aimed to create a membrane environment as physiologically relevant as possible, by following key general tendencies observed in specific brain post-mortem studies relevant for the adenosine A$_{2A}$R receptor (A$_{2A}$R) biology[63–68]. The CHARMM-GUI membrane builder[61] was used to build a multicomponent membrane by using key representative lipids (that is, polyunsaturated phospholipids, cholesterol or sphingomyelin) while keeping an adequate balance between all components. Thus, our membrane was made of cholesterol, saturated phospholipids, mono- and polyunsaturated phospholipids and sphingomyelin (Supplementary Table 2). As described in the main manuscript, one A$_{2A}$R receptor was then embedded into this membrane using VMD 1.9 (ref. 62) and the system was then solvated, neutralized and the ionic strength was adjusted using the CHARMM-GUI membrane builder[61]. Thereafter, a VMD script was used to re-hydrate the membrane patch using $\sim 30$ water molecules (TIP3P model) per lipid. The exact composition of the simulation systems is detailed in Supplementary Table 1. The lipid composition used yielded a protein-to-lipid ratio of 1:337.

*Simulation protocol.* As described in the Methods section of the main manuscript, simulation sets 1 and 2 (Supplementary Table 1) share a common building phase, whereas the set-up of simulation sets 3 and 4 involved a complete substitution of the membrane environment or the inclusion of the ICL3, respectively. At the beginning of the equilibration phase for simulations sets 1 and 2 (Supplementary Table 2), harmonic positional restraints were applied to the C$_\alpha$ atoms of the protein and the system was simulated for 10 ns. Such constraints were gradually released from the receptor over 5 ns and the system was further equilibrated for 100 ns. In simulations set 3 (that is, POPC), harmonic positional restraints were applied to all atoms of both protein and cholesterol and the system was equilibrated for 20 ns in the NPT ensemble. In simulation set 4 (that is, ICL3 included), after 10,000 steps of minimization, a gradual release of different applied harmonic constraints was used during 40 ns phase in the NPT ensemble. Harmonic restraints were first applied during 10 ns to all atoms of the system except for the intracellular polar head region of all membrane lipids. Constraints were then

released for all water and ion atoms and the system was simulated for another 10 ns. Subsequently, the ICL3 region was released and the system (that is, only protein and target cholesterol restrained) was further equilibrated for 20 ns. NPT simulations were carried out at 310 K and 1 bar using the Berendsen barostat[69] with a relaxation time of 400 and 2 fs integration time step. NVT simulations were run at 310 K, using the Langevin thermostat[70] with a damping coefficient of 5 ps$^{-1}$ and 4 fs integration time step. All along the simulations, van der Waals and short-range electrostatic interactions were cut off at 9 Å and the particle mesh Ewald method[71] was used to compute the long-range electrostatic interactions.

*Inclusion of ICL3.* The ICL3 structure was taken from PDB:3PWH and included in the A$_{2A}$R structure using MODELLERv9.10 (ref. 72) and VMD[62]. The system was first minimized and the ICL3 subsequently equilibrated and relaxed for 50 ns in the NPT ensemble (see simulation protocol above).

**In silico model and conformational analysis for MTSEA-B chemical modification.** In a first step, the side chain of cysteines (C3.30, C5.46 and C6.56) of the A$_{2A}$R (PDB:3EML) was chemically modified by attaching MTSEA-B using the builder tool of the MOE package (version 2016.08). In a second step, the conformational space of such chemical modification was explored using the LowModeMD method in the MOE package with: Rejection Limit 100, Iteration Limit 100, RMS Gradient 0.1, MM Iteration Limit 500, RMSD Limit 0.5, Energy Window 100, Conformation Limit 1000 and applying the Amber10:EHT force field. As a result of this search, we obtained 98 different conformers for the chemically modified residues C3.30 and C5.46 and 96 conformers for the chemically modified residue C6.56, as shown in Supplementary Fig. 15.

**Umbrella sampling simulations and free-energy calculations.** Biased simulations were used to compute the free energy of cholesterol extraction from the inside of the A$_{2A}$R. To this end, we used the simulation engine Gromacs v5 (ref. 73) in combination with the CHARMM36 force field[58]. The recommended settings including the usage of Verlet lists, PME with 1.2 nm cutoff for electrostatics and a cutoff function for the Van de Waals at 1.2 nm with a force-switch starting at 1.0 nm were used[74]. The composition of the system is detailed in Supplementary Table 3. The potential of mean force was computed along the direction perpendicular to the membrane (that is, $z$ axis). No bias was introduced at the $xy$ axis, that is, molecules could freely diffuse in the membrane plane ($xy$). The biased reaction coordinate was defined as the distance along the $z$ axis between the centre of mass of the A$_{2A}$R backbone and the oxygen atom of cholesterol. In total, 35 different windows were probed. Harmonic restraints with a force constant of $2,000\,\text{kJ}\cdot\text{mol}^{-1}\cdot\text{nm}^2$ were imposed along the reaction coordinate. To ensure an adequate overlap between neighbouring windows, each window is separated by 0.1 nm along the reaction coordinate. Based on the quality of the data and the presence of large energy barriers, each window was simulated within 200–600 ns. The accumulated simulation exceeds 8 μs. The reaction coordinate was monitored every 10 fs within each window. The last frame of the 200 ns simulation of the previous window was used as the starting configuration of the next one. Although this protocol cannot be used to simulate all windows in parallel, it drastically improves the equilibration time for each window. The first window corresponds to the cholesterol deeply buried in the ligand cavity. In the last windows, cholesterol moves freely in the water solution. We used the g_wham tool distributed with gromacs v5 (ref. 73) to compute the PMF (Supplementary Fig. 16) disregarding the first 50 ns of each trajectory to avoid sampling potential non-equilibrium configurations. Autocorrelation time of the data was used to minimize correlation effects and to select the appropriated gathering frequency in the PMF curves. Bootstrapping of 1,000 samples was used to estimate the standard deviation of the PMF results.

**Tunnel pathway calculations.** Tunnel pathways were computed using the Caver software[49]. The starting point coordinates were set to centre of the binding pocket of opsin and A$_{2A}$R. Computations were carried out using a shell radius 3 Å, shell depth 4 Å, a probe radius of 1.1 Å for the opsin receptor and 1.4 Å for the A$_{2A}$R. Obtained results were plotted using the VMD software.

**Data availability.** The authors declare that all data necessary to support the findings of this study are available within the paper and its Supplementary Information Files. Additional data that were omitted from the paper are available from the corresponding authors upon reasonable request. Crystal structure coordinates from the Protein Data Bank (3EML, 3PWH, 4EIY) were used in this study.

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

## Acknowledgements

J.S. and R.G.-G. acknowledge support from Fundació La Marató de TV3 (091010), Instituto de Salud Carlos III FEDER (CP12/03139 and PI15/00460), and the GLISTEN European Research Network. F.S., M.P., J.S. and R.G.-G. acknowledge support from Ministerio de Educación y Ciencia (Grant Number: SAF2009-13609-C04-04) and H2020 Project MedBioinformatics under Grant Agreement Number 634143. P.W.H. and R.G.-G. acknowledge support from the Deutsche Forschungsgemeinschaft (DFG HI 1502/ 1-1, WO 1908/2-1, SFB 740) SFB70 and the Norddeutscher Verbund für Hoch- und Höchstleistungsrechner (HLRN). M.Mar. and J.L.A. acknowledge support from Ministerio de Economía y Competitividad (BFU2011–23034) and JCCM (PEII-2014-030-P). I.R.-E. acknowledges support from AGAUR and European Social Fund (2015 FI_B00145). H.M.-S. and M. Man acknowledge financial support from the Academy of Finland and thank the CSC—IT Center for Science (Espoo, Finland) for providing computational resources.

## Author contributions

R.G.-G. designed and performed the classical MD simulations with contribution from J.S., I.R.-E. and P.W.H. M.Man. and H.M.-S. performed the umbrella sampling simulations. H.M.-S. performed the free energy calculations. J.L.A. and M.Mar. performed cholesterol and MTSEA-B experiments. J.S. performed tunnel pathway calculations. P.W.H., M.P. and F.S. gave technical support and conceptual advice. R.G.G. and J.S. wrote the paper with contributions from J.-L.A., M.Mar., M.M.-S., H.M.-S, and comments from all authors.

## Additional information

**Competing financial interests:** The authors declare no competing financial interests.

