## [Peer Review File · Nature Communications]

Reviewers' comments:

Reviewer #1 (Remarks to the Author):

This manuscript presents combined experimental and computational study of effects of cholesterol on ligand binding to A2A GPCR. Computationally, using extensive set of simulations, the authors identify specific binding mode of cholesterol inside the ligand binding pocket of A2A. These results are used to explain mechanistically how elevated levels of cholesterol inhibits ligand binding as measured in experiments. Furthermore, to provide experimental validation for the specific binding mode of cholesterol found in the simulations, additional biotinylation experiments were carried. The topic of membrane effects on GPCR function is generally very important and I find the entire study interesting and the manuscript well constructed. However, I have some concerns about how experimental results in the manuscript were interpreted which may have an overall impact on the formulated mechanism of how cholesterol affects ligand binding.

My first concern is regarding Figures 1 and 2 and conditions used for the corresponding experiments. Specifically, in Figure 1 time course is shown of specific ligand binding as cholesterol is gradually removed from the cells. My concern is that these experiments were conducted under one particular concentration of the ligand. And yet, the results are interpreted as cholesterol having inhibitory effect. But I am wondering how the results would change if for particular cholesterol levels one considers range of ligand concentrations. The same issue is with Figure 2 in which only single conditions are probed. I believe that it is important to investigate the effect of saturating ligand concentrations. This will enable more proper comparison of conditions with different cholesterol contents.

I also have a concern about how the biotinylation experiments were carried out. A2A has multiple endogenous Cys residues and therefore, for the experiment to be interpretable, all the Cys residues except the one that is of interest must be first mutated. Otherwise, the observed effect can be coming not necessarily from Cys in the binding site but from other Cys. Since I do not find any indication in the Methods section that indeed Cys-less A2A was considered (apart from Cys of interest, there also could be some Cys residues that are involved in disulfide bonding), I assume that this was not done. As such, the interpretation of the results presented in Figure 6 may not be correct.

I also have a comment about computational part in relation to the role of different lipids for cholesterol penetration. The simulations show that in 30% simulations with original (more physiological) lipid mixture cholesterol advanced towards the receptor interior. But it did not in 70% of such simulations. This statistics is compared to another set of simulations carried out in POPC membranes. In these systems cholesterol backs away in 50% of simulations. I believe that these percentages alone cannot be conclusive to say that cholesterol dynamics in one system is different than in the other one. So I think more quantification of this argument is required or this conclusion should be softened.

Reviewer #2 (Remarks to the Author):

A Summary of the key results

The key result of the article "Membrane cholesterol can access the interior of G protein-coupled receptors: the adenosine A2A receptor case" is the description of an effect of cholesterol on adenosine A2aR binding that is not purely allosteric, but seems competitive at the orthosteric site.

B Originality and interest: if not novel, please give references

This is an original observation that, potentially, could have important implications for all class of GPCRs.

C Data & methodology: validity of approach, quality of data, quality of presentation
The binding approach needs to be revised in accord to what suggested in (F)

D Appropriate use of statistics and treatment of uncertainties

Statistic is appropriate for binding data presented, except that in the Figure legends, Authors should specify how many replicates were done for each experiment, it is not enough to indicate "from, at least 3 separate experiments".

E Conclusions: robustness, validity, reliability

The robustness of the conclusions is biased by the fact that at this stage results could be interpreted alternatively, with M β CD and WSC altering the turnover of the A2aR in intact cells (see F).

F Suggested improvements: experiments, data for possible revision

This referee feels that additional binding experiments are needed to strengthen the results and exclude possible artifacts.

A list of them is summarized below.

1) From the graph of Figure 1 it is evident that at time longer than 40 minutes M β CD decreases the binding of [3H]ZM241385. The Authors assume that the decrease in binding from 50 to 240 minutes follows the interpolation. Nevertheless, this assumption is risky; binding could fall very rapidly, as suggested by its trend between 40 and 50 minutes. This is not a detail, because in the experiment of Figure 2 cells are exposed to M β CD + WSC for 90 minutes. The Authors should clearly specify whether the compounds were used in sequence (M β CD - than washing - than WSC), or they were used together in the bath for 90 minutes. In the latter case, additional time points should be added to the time course of Figure 1. Exposure of the cells to M β CD for 90 minutes, could decrease the binding of [3H]ZM241385 to a level that could make data difficult to interpret. The concentration of M β CD utilized in Fig. 1 is not reported anywhere. Is it the same utilized in Figure S1?

2) Many membrane proteins undergoes to a continuous turnover of internalization and recycling (Hao and Maxfield, Characterization of rapid membrane internalization and recycling. *J Biol Chem.* 2000 May 19; 275(20):15279-86). If internalization is inhibited the number of receptors on the membrane increase. Cholesterol is a main component of caveolae and less of clathrin coated pits, and its depletion inhibits receptor internalization (Agathe et al., 1999 Acute cholesterol depletion inhibits clathrin-coated pit budding. *Proc Natl Acad Sci U S A.* 1999 Jun 8; 96(12): 6775-6780). A2aR are internalized through both, caveolae and clathrin coated pits and they undergo to constitutive internalization and recycling (Klaasse et al. Internalization and desensitization of adenosine receptors. *Purinergic Signal.* 2008 Mar; 4(1): 21-37; Burgueño et al., The adenosine A2A receptor interacts with the actin-binding protein alpha-actinin. *J Biol Chem.* 2003 Sep 26; 278(39):37545-52). Given that, the Authors should exclude that the effect of M β CD is due to inhibition of A2aR internalization, and redistribution of the receptors on the plasma-membrane.

3) If cholesterol is a competitive ligand at the orthosteric site, high concentrations of WSC should completely displace [3H]ZM241385 from the A2aR binding pocket. To address this point, a competition binding curve of WSC against [3H]ZM241385 should be done. If [3H]ZM241385 is not completely displaced, it is hard to believe that cholesterol bind to the orthosteric site, unless we assume that the orthosteric binding pocket can accommodate both cholesterol and 3H]ZM241385. Nevertheless, this can be excluded by the conclusion of the Authors "our data indicate that, if cholesterol reaches the interior of the protein, it will likely hamper the binding of A2AR ligands such as ZM241385 to the orthosteric binding site." Furthermore, a competition binding curve will consent to measure the affinity of cholesterol for the orthosteric site. Such an experiment should be done on membrane preparations, in order to avoid the confounding

effect of other mechanisms that could be induced by WSC treatment in intact cells (receptor internalization for instance).

4) Figure S3 shows an inhibitory effect of WSC on filipin III staining. It is important to have a control with cells that do not express A2aR to confirm that the effect of filipin III is specific for cells that express the A2aR.

G References: appropriate credit to previous work?
References are appropriate.

H. Clarity and context: lucidity of abstract/summary, appropriateness of abstract, introduction and conclusions

Abstract, introduction and conclusions are appropriate pending the exclusion of the alternative interpretation of the data (see F).

Reviewer #3 (Remarks to the Author)

The manuscript "Membrane cholesterol can access the interior of G-protein coupled receptors: The adenosine A_{2A} receptor case" makes the bold claim that cholesterol enters the orthosteric binding pocket of A_{2A} and competes for ligand binding. While this is certainly an intriguing hypothesis, it is also a very surprising one, and therefore requires a very high standard of proof. Unfortunately, the data do not reach this standard, and I therefore do not recommend publication as a Nature Communication. Below I explain first why this is surprising, and then second why the data do not support the conclusion stated in the title.

Results contradict published reports of cholesterol-dependent GPCR function.

The authors report in Fig. 1 that radioligand binding is *enhanced* upon cholesterol *depletion* with MβCD. All published data of which I am aware indicate the opposite — that the presence of cholesterol rather enhances or is even necessary for GPCR ligand binding and function. The authors cite some of this work, but do not mention that it contradicts their own conclusion. For example, the authors cite O'Malley et al's work (ref 21) in which A2A is reconstituted into detergent micelles, and in which it is found that that the receptor does *not* bind ligand unless a sterol derivative (cholesterol hemisuccinate, in this case) is present in sufficient quantity. The present manuscript use the O'Malley result to support their simulation results, in particular that they find a similar number of cholesterol molecules in the vicinity of the receptor. However, they fail to mention that the O'Malley data contradict their own conclusions.

The authors also mention other reports of cholesterol mediated GPCR function (refs 18 and 19), stating that "Our finding goes along with previous experimental evidence in various class A GPCRs pointing towards a cholesterol-mediated modulation of ligand binding." However, they fail to mention that refs 18 and 19 also contradict their own conclusion. This is a bit disingenuous, to put it mildly. Lastly, rhodopsin clearly requires significant cholesterol to function, as the rod outer cell membrane contains ~ 40 mol% cholesterol.

It is therefore incumbent upon the authors to discuss and explain these discrepancies, and to explain why their experimental protocol yields different results compared to previously published data. One potential source of discrepancy may be in the interpretation of the cholesterol depletion experiments.

Cholesterol depletion and cholesterol content assay

The authors state that treatment of the cells with 5 mM M β CD for 30 min depletes “most of the cholesterol from the cell membranes.” Though the authors are not explicit about what is meant by “most”, one assumes that it must mean at least 51%. (Note that visual inspection of Fig. 1 is not sufficient to answer this question, as the data indicate some unexplained variability of cholesterol content with time of M β CD treatment. Why does cholesterol content increase from 30 to 40 min of M β CD treatment?) This level of depletion is not supported by reports in the literature, for example Mahammad and Parmryd, BBA 1778: 1251(2008), which carefully assayed cholesterol removal from membranes by M β CD, and found at most 35% reduction in membrane cholesterol levels. (Note that these authors also report that M β CD is promiscuous — it removes lipids other than cholesterol from the membrane.)

Cholesterol is then reintroduced after depletion using water soluble cholesterol, and measured with a different assay (Filipin staining), on the basis of which it is stated that cholesterol has recovered to some extent. These data are presented in the SI, and are not convincing, as they are not a direct measure of cholesterol content and are not easily compared to the initial assay.

Overall, the data on cholesterol content after depletion and following reintroduction using water soluble cholesterol need to be corroborated, e.g., by lipidomic analysis.

Lastly, after 30 min of M β CD treatment, the cells are at the least very stressed, and most likely rapidly dying. Thus other mechanisms may well be responsible for the observed reduction in ligand binding upon M β CD treatment. Perhaps less protein is available at the surface, for example?

Molecular simulation results

The structural explanation for the biochemical results is obtained on the basis of molecular simulations, but there are also some significant questions regarding the simulation protocol and therefore the conclusions.

(1) Intracellular loop 3 (between helix 6 and 7) is not included in the simulation. According to the SI at the top of the first column on page 11, the authors indicate that this loop is not included in the simulation, as it was not resolved in the crystal structure. Given that the conformation of helix 6 must undergo significant displacement to allow the cholesterol to enter (it enters between helices 5 and 6), this is a significant omission. Why did the authors not perform additional simulations that include the loop? Given the simulation time devoted to the present results, it would be a simple matter to build in the loop (perhaps from one of the Tate lab/Heptares structures) and relax it with a few microseconds of unbiased MD.

(2) The cholesterol entrance event was observed in 1 out of 4 simulations. Configurations were then taken from this one trajectory and used to seed other simulations, ostensibly as some sort of statistical test. However, this does not provide any information on whether the results are reproducible, since only the “interesting” event is considered in the follow up. The authors also perform umbrella sampling in an effort to determine the cost of extracting cholesterol from the binding pocket via the aqueous phase. Why did they not perform this calculation for the entrance/exit of cholesterol via the membrane? Wouldn't this be an obvious way to corroborate the single event observed in the unbiased simulation?

(3) At the top of the second column on page 5, the authors speculate about the role of the membrane environment in driving cholesterol into the receptor. The simulation in which cholesterol enters the receptor includes a rather complex membrane environment, with 5 different lipids plus cholesterol, including PUFA and sphingolipid. Based on a comparison to a simulation in

chol/POPC bilayer, they conclude that in the a "an unsaturated lipid along with a cluster of 4 cholesterol molecules seem to assist cholesterol entering the receptor..." Needless to say, these are anecdotal at best, and are not sufficient to support the statement (at the end of the paragraph) that the membrane environment assists in the cholesterol entry. I am also not certain at all whether the complex membrane is "thicker and more compact" than the POPC/chol control. The complex membrane has saturated chains and cholesterol, but it also has a significant amount of PUFA.

Response to reviewers

Reviewer#1 (R1)

Remarks to the Author:

This manuscript presents combined experimental and computational study of effects of cholesterol on ligand binding to A_{2A} GPCR. Computationally, using extensive set of simulations, the authors identify specific binding mode of cholesterol inside the ligand binding pocket of A_{2A}. These results are used to explain mechanistically how elevated levels of cholesterol inhibits ligand binding as measured in experiments. Furthermore, to provide experimental validation for the specific binding mode of cholesterol found in the simulations, additional biotinylation experiments were carried.

The topic of membrane effects on GPCR function is generally very important and I find the entire study interesting and the manuscript well constructed. However, I have some concerns about how experimental results in the manuscript were interpreted which may have an overall impact on the formulated mechanism of how cholesterol affects ligand binding.

We thank Reviewer 1 for carefully reviewing our manuscript and providing interesting feedback, concerns and suggestions. We have addressed point-by-point all comments from the reviewer and improved the manuscript accordingly (see below).

R1.1) *My first concern is regarding Figures 1 and 2 and conditions used for the corresponding experiments. Specifically, in Figure 1 time course is shown of specific ligand binding as cholesterol is gradually removed from the cells. My concern is that these experiments were conducted under one particular concentration of the ligand. And yet, the results are interpreted as cholesterol having inhibitory effect. But I am wondering how the results would change if for particular cholesterol levels one considers range of ligand concentrations.*

The same issue is with Figure 2 in which only single conditions are probed. I believe that it is important to investigate the effect of saturating ligand concentrations. This will enable more proper comparison of conditions with different cholesterol contents.

We completely agree with Reviewer 1 that in order to confirm an increase in A_{2A}R specific binding upon cholesterol depletion (Figs. 1 and 2), a range of ligand concentrations including a saturating one need to be probed. Thanks to this suggestion, we have significantly improved the manuscript with specific changes and new experiments (please see below).

1. First, we would like to highlight that experiments shown in Figs. 1 and 2 were indeed carried out using a saturating concentration of the radioligand [3H]ZM241385 (i.e. 40 nM). We assume 40 nM to be a saturating concentration based on previous work of our lab¹, where we characterized the saturation binding profiles of different adenosine receptor types including the A_{2A}R using identical experimental conditions. Specifically, in this study¹ we probed a wide range of concentrations (i.e. from 1 to 20 nM) to build the saturation binding profile of the [3H]ZM241385 ligand at the A_{2A}R in intact C6 glioma cells. Since this was not

clearly stated in the manuscript, we now include this detail in the caption of Figs. 1, 2 and 6 as well as in the Methods section, see below:

Fig. 1, caption

“...**binding to A₂AR in intact cells and membrane cholesterol content. This experiment was carried out using a saturating radioligand concentration of 40 nM. Mean ± SEM values...**”

Fig. 2, caption

“...for 40, 50 or 90 minutes. **These experiments were carried out under saturating radioligand concentration (i.e. 40 nM). These results are m**Mean ± SEM...”

Fig. 6, caption

“(A) Effect of the MTSEA-B reagent and cholesterol depletion on [3H]ZM241385 specific binding **at a saturating radioligand concentration of 40 nM.** (B) Scheme of cholesterol...”

2. In addition, we fully agree with the reviewer that probing a range of ligand concentrations would consistently validate the inhibitory effect of cholesterol on ligand binding. Therefore, we carried out a comparative saturation binding assay in both control cells (i.e. untreated) and cells depleted from cholesterol (i.e. treated with 5 mM MβCD for 40 min). These new results further confirm the increase in A_{2A}R specific binding upon cholesterol depletion shown in Figs. 1 and 2, thus validating the inhibitory role of cholesterol. We now show the data from the comparative saturation binding experiments in Fig. S5 and have improved the manuscript to reflect this validation as follows:

Main text

“Remarkably, the lack of cholesterol increases the specific binding of [3H]ZM241385 to the A_{2A}R by more than 100 % (30 min) when compared to non-treated ~~intact~~ ~~cells~~ ~~membranes~~ (0 min) (Fig. 1 and Fig. S1). **Saturation binding experiments using a wide range of radioligand concentrations confirm this inhibitory effect (see Fig. S5).** To validate the reversibility of this effect, we...” (Page 3, paragraph 3, lines 1-5)

Fig. S5

“**Figure S5 | Effect of MβCD on specific A_{2A}R binding in C6 intact cells.** Control (closed circles) and 5 mM MβCD (40 minutes) (open circles) treated cells were incubated with different concentrations of [3H]ZM241385 as described in the Methods. These results are mean ± SEM values obtained from six separate experiments carried out in duplicate. Kinetic parameters (B_{max} and K_d) of the corresponding saturation binding curves are indicated at the bottom the figure. ****p < 0.01 significantly different from control value.**”

Methods

“...Intact C6 glioma cells were incubated with 40 nM [3H]ZM241385 (i.e. **saturating ligand concentration, based in our previous work⁵⁰**) for 2 h at 25 °C in the presence of 5 U/ml ADA in order to remove endogenous adenosine. **A range of 1.25-40 nM [3H]ZM241385 was employed in saturation binding assays.** Non-specific binding was...” (Page 11, paragraph 1, lines 32-38)

R1.2) *I also have a concern about how the biotinylation experiments were carried out. A_{2A}R has multiple endogenous Cys residues and therefore, for the experiment to be interpretable, all the Cys residues except the one that is of interest must be first mutated. Otherwise, the observed effect can be coming not necessarily from Cys in the binding site but from other Cys. Since I do not find any indication in the Methods section that indeed Cys-less A_{2A}R was considered (apart from Cys of interest, there also could be some Cys residues that are involved in disulfide bonding), I assume that this was not done. As such, the interpretation of the results presented in Figure 6 may not be correct.*

Reviewer 1 raised here an extremely interesting aspect of the biotinylation experiments used in this manuscript. We completely agree that not only one but rather different A_{2A}R endogenous cysteines (both inside and outside the receptor) are potentially susceptible to biotinylation (point **2**). We also agree that our experimental set-up is not sufficient to pinpoint which particular cysteine residue/s is behind the cholesterol effect (see point **3**). Nevertheless, based on different structural (point **2**) and methodological (point **1**) aspects we detail below, we can confidently state that regardless of the specific cysteine residue involved cholesterol, cholesterol will necessarily need to access the interior of the A_{2A}R to exert the observed effect.

- 1. Methodological aspects:** two key aspects of the biotinylation method need to be taken into account first:
 - a. The basis of the method we employ in this manuscript is the reaction of methanethiosulfonate (MTS) reagents with the sulfhydryl group of accessible cysteine residues². Thus, only cysteines at the water-accessible surface of the receptor, namely the extracellular side or interior of the membrane-spanning segment, will be reactive to MTS reagents (i.e. cysteine residues at the lipid-accessible surface will not be reactive).
 - b. MTS reagents are specific for free sulfhydryl groups, and thus residues engaged in a disulfide bridge are not reactive even if they are accessible to water^{2,3}.
- 2. Structural aspects:** as pointed out by reviewer 2, different A_{2A}R endogenous cysteines are likely water-accessible, and hence potentially susceptible to biotinylation. Nevertheless, as shown by high-resolution crystallography data⁴, water-accessible cysteine residues with free sulfhydryl groups are only found in the interior of this receptor whereas extracellular water-accessible cysteines are all engaged in disulfide bonds and therefore cannot react with the biotinylation reagent (see point **1b** above and detailed description in the new Section S3). Therefore, we can confidently state that, regardless of the cysteine residue involved, cholesterol will need to access the receptor interior in order to exert the described effect. We would like to highlight that this finding is the first evidence for a completely new mechanism on how cholesterol could control GPCR functionality.
- 3. Challenges and limitations:** we strongly agree with the reviewer that our experimental set-up is not sufficient to discriminate which particular cysteine residue is behind the observed effect. We believe these limitations are not

adequately addressed in the manuscript. Accordingly, we have changed the text to reflect two key aspects as follows:

- a) Residue C3.30 is the most likely candidate for biotinylation out of those cysteines susceptible to reaction with MTSEA-B at the water-accessible surface of the protein interior. Nevertheless, this residue is not the only candidate, as we now thoroughly describe this in section S3 of the SI.
- b) The exact structural region sampled by cholesterol once inside the receptor cannot be described from our experiments but rather suggested by our comprehensive state-of-the-art simulations along with previous spectroscopy⁵ and crystallography⁶ experiments demonstrating a similar accessibility and occupancy of the ligand binding pocket of rhodopsin.

All in all, whilst the main finding of our work remains (i.e. cholesterol can access the interior of the A_{2A}R), we now discuss this issue more critically and have corrected any misleading message regarding the contribution of specific cysteines to the biotinylation/cholesterol effect:

Abstract

“...opsin ligands. Moreover, chemical modification of the A_{2A}R ~~interior binding pocket~~ via biotinylation confirms...”

Introduction

“...approach to assess cholesterol impact on chemical modification of the A_{2A}R ~~interior orthosteric binding pocket~~. Taken together, our...” (Page 3, paragraph 1, lines 1-2)

Results

Biotinylation experiments confirm that cholesterol can occupy the interior of the A_{2A}R ~~orthosteric binding site~~. Providing experimental...” (Page 7, paragraph 3, line 1)

“...cholesterol ability to occupy the ~~interior orthosteric binding pocket~~ of this receptor. In this study...” (Page 7, paragraph 3, lines 9-10)

“...as derivatives of methanethiosulfonate (MTS) (~~detailed in section S3 of the SI~~). As shown by the A_{2A}R crystal structure (PDB: 3EML)²³, ~~water-accessible~~ cysteine residues ~~C3.30 provides the only thiol group exposed to the binding crevice~~ with a free sulfhydryl group are only found in the interior of the receptor (see Fig. S17 and Table S5). In our experiments, we chemically modified by biotinylation ~~this~~ these water-exposed cysteines using...” (Page 7, paragraph 3, lines 20-28)

“...Fig. 5C illustrates that ~~a-biotinylation of~~ residue C3.30 (green sticks) ~~results in a modified side chain that occupies~~ a large region of the orthosteric binding site...and red sticks, respectively Fig. 5B). ~~As shown in Fig. S18, chemically modified cysteine residues C5.46 and C6.56 yield a similar overlap with the ZM241385 ligand.~~ Accordingly, if cholesterol is able to invade the A_{2A}R binding pocket as proposed by our simulations, ~~C3.30 cysteine~~ biotinylation should be...” (Page 8, paragraph 1, lines 2-13)

“...using MTSEA-B to covalently modify ~~cysteine residues in the A_{2A}R interior C3.30~~ and MβCD to deplete...” (Page 8, paragraph 2, lines 3-5)

“...compared to untreated conditions (Bar 1). Then, we assessed if ~~cysteine residues in the A_{2A}R interior C3.30 is~~ are susceptible to MTSEA-B...” (Page 8, paragraph 2, lines 13-14)

“This marked reduction in specific binding strongly indicates that ~~C3.30~~ at least one of the cysteine residue in the A_{2A}R interior (see section S3) is susceptible to biotinylation...we tested the influence of ~~orthosteric~~ cholesterol binding in ~~cysteineC3.30~~ biotinylation (Bar 4)...has a shielding effect by protecting cysteine residues from biotinylation ~~inside the receptor~~~~at the orthosteric binding site~~. Thus, this observation supports the presence of a cholesterol molecule ~~inside the A_{2A}R transmembrane bundle~~ ~~the orthosteric binding site~~ with the ability to block access of MTSEA-B to ~~water-accessible cysteine residues~~~~C3.30~~. Finally,...of cholesterol and biotinylation ~~on binding site occupancy~~ that hampers binding of [³H]ZM241385, yielding...” (Page 9, paragraph 1, lines 1-9, 13-19, and 22-24)

Discussion

“...be involved in reducing ZM241385 binding. However, our work reveals an ~~additional~~~~different~~ and unexpected mode of cholesterol action. We have observed for the first time that cholesterol ~~modulates~~ ~~competes for~~ orthosteric ~~ligand binding~~ ~~deep~~ ~~inside of~~ at the A_{2A}R after entering from...” (Page 10, paragraph 1, lines 9-14)

Methods

“**Water-accessible cysteine residues in the A_{2A}R.** According to high-resolution crystallography data³⁸, water-accessible cysteine residues with free sulfhydryl groups are only found in the interior of the receptor as all extracellular water-accessible cysteines are engaged in disulfide bonds and therefore cannot react with the biotinylation reagent (see detailed description in section S3 of the SI). Among these residues, C3.30 is the closest one to the ZMA ligand and is therefore used as representative in this manuscript (see Fig. 5C).” (Page 12, paragraph 2, lines 1-11)

Apart from changing the manuscript, and to provide the reader with an adequate overview of this issue, we have substantially improved the SI by creating two figures, one table and a whole new section (i.e. section S3) with a detailed structural characterization of putative A_{2A}R cysteine residues.

SI

“**Section S3 - Biotinylation experiments**

1. Details about the methodology: originally, the Substituted-Cysteine Accessibility Method (SCAM) was developed to elucidate water-accessible residues in membrane-spanning proteins like channels³¹, transporters³² or binding-site crevices³³. In the absence of crystallography data, this method requires a systematic mutation of every protein residue into a cysteine followed by an assessment of ligand binding properties. Throughout a decade, the SCAM method has been employed by Javitch et al. to explore water-accessible residues in one G protein coupled receptor, namely the dopamine D₂ receptor (D₂R)³³⁻³⁵. As mentioned in this work, an ideal starting point would be to create a cysteine-free pseudo-wild-type background that is insensitive to the reagents and has normal expression and function. However, intense efforts failed to achieve such construct for the D₂R³⁶. It is likely that the lack of cysteine residues in cysteine-free pseudo wild type GPCRs impacts protein expression, folding and thus the formation of a functional receptor.

In the present study, we adapted the SCAM method to explore the reactivity of cysteine residues under different conditions in the A_{2A}R interior. The basis of the employed methodology relies on two key aspects:

- a. Methanethiosulfonate (MTS) reagents react with sulfhydryl groups of accessible cysteine residues³⁶. Thus, only cysteines at the water-accessible

surface of the receptor, namely the extracellular side or interior of the membrane-spanning segment, will be reactive to MTS reagents (i.e. cysteine residues at the lipid-accessible surface will not be reactive).

- b. MTS reagents are specific for free sulfhydryl groups, and thus residues engaged in a disulfide bridge are not reactive even if they are accessible to water^{36,37}.

In contrast to the work from Javitch et al., in the present study we know the exact location of water-accessible cysteines in the A_{2A}R thanks to the availability of high resolution crystallography data (PDB:3EML). This knowledge is crucial for correct interpretation of biotinylation and binding experiments as outlined below.

2. **Ability of cysteine residues to react with MTSEA-B in the A_{2A}R.** In order to assess the ability of the A_{2A}R to react with the biotinylation reagent MTSEA-B, the receptor was scanned for cysteine residues, their location and engagement in disulfide bridges. As shown by high resolution crystallography data (PDB: 3EML) (Fig. S17, Table S5), 8 out of 14 endogenous cysteines residues are engaged in 4 disulfide bridges at the extracellular side of the receptor, and thus these 8 residues are not susceptible to be biotinylated (see point **1b** above). The 6 remaining cysteines are located in the transmembrane portion of the protein, and thus they can only face either the membrane (2 cysteines) or the interior of the receptor (4 cysteines). Since lipid-accessible cysteine residues are not reactive (see point **1a** above), there are only 4 cysteines in the receptor interior which can potentially react with the biotinylation reagent (C3.30, C4.49, C5.46, C6.56) (Fig. S17, Table S5). A structural inspection of those cysteines shows that C4.49 is buried between TM3 and TM4 (i.e. not water-accessible) yielding only 3 reactive cysteines in the receptor interior that are susceptible to be biotinylated, namely C3.30, C5.46, and C6.56. As shown in Fig. S18, chemical modification of these cysteine residues by MTSEA-B would clearly overlap with the orthosteric binding site of the A_{2A}R.

In summary, a detailed structural characterization of the A_{2A}R suggests that all cysteines outside the receptor are non-reactive as they are engaged in disulfide bridges and that only cysteines inside the receptor are reactive towards the biotinylation reagent MTSEA-B. Hence, regardless of which cysteines become biotinylated in the receptor interior, cholesterol needs to enter the receptor interior to exert its action. It is worth noting that based on the experimental set-up used in this paper, we cannot pinpoint the exact cysteine residue reacting with the biotinylation reagent. However, provided that MTSEA-B should access the receptor from the extracellular side, we can speculate that the first cysteine residue (i.e. C3.30) found on the MTSEA-B entrance pathway into receptor should mostly react with the biotinylation reagent.” (SI: Pages 4-5)

Fig. S17

“**Figure S17 | Location of cysteine residues in the A_{2A}R and accessibility to chemical modification via MTSEA-B.** Cysteine residues (liquorice representation) susceptible to biotinylation in the A_{2A}R (grey cartoons) are shown in yellow transparent surface whereas non-reactive residues are shown in grey transparent surface. The antagonist ZM241385 is shown in red liquorice. The PDB:3EML structure was used to generate this figure.”

Fig. S18

“**Figure S18 | In silico model of MTSEA-B chemical modification of A_{2A}R cysteine residues.** Chemical modification of the three cysteines (C3.30, C5.46, C6.56) in the binding pocket shows that the covalently bound biotin (green surface) competes with other orthosteric ligands such as ZMA (red stick representation) in the binding pocket.”

Table S5

Table S5 | Location and water-accessibility A_{2A}R cysteine residues*

Residue ID	Location	Water-accessible
C3.30	TM3	Yes
C5.46	TM5	Yes
C6.56	TM6	Yes
C1.54	Lower TM1	No
C4.49	Lower TM4	No
C6.47	TM6	No
C71	EC**	No
C74	EC**	No
C76	EC**	No
C146	EC**	No
C159	EC**	No
C166	EC**	No
C259	EC**	No
C262	EC**	No

* **Water-accessible cysteines highlighted in bold**

** **EC: extracellular loops (locked in disulfide bridge)**

R1.3) *I also have a comment about computational part in relation to the role of different lipids for cholesterol penetration. The simulations show that in 30% simulations with original (more physiological) lipid mixture cholesterol advanced towards the receptor interior. But it did not in 70% of such simulations. This statistics is compared to another set of simulations carried out in POPC membranes. In these systems cholesterol backs away in 50% of simulations. I believe that these percentages alone cannot be conclusive to say that cholesterol dynamics in one system is different than in the other one. So I think more quantification of this argument is required or this conclusion should be softened.*

We agree with the reviewer that a better quantitative description should be employed here in order to compare both systems. Therefore, instead of using such percentages we now provide concrete statistical values (i.e. mean and standard error of the mean) of the distance used to measure cholesterol progressing towards the interior of the receptor for each set of 10 x 100ns simulations. Thanks to the suggestion of the reviewer, we now clearly show that the membrane environment (i.e. original more physiological versus POPC) significantly affects the entrance and progression of cholesterol towards the interior of the protein.

To reflect these changes, we have created a new table in the Supplementary Information (Table S4) and have amended the text of the main manuscript accordingly:

Main text

“...to enter the receptor. ~~Overall, cholesterol significantly advances towards the receptor interior in 30% of the trajectories (i.e. 12 out of 40 trajectories)~~ These data clearly suggest...” (Page 6, paragraph 1)

“Interestingly, in the absence of a more compact and thicker membrane, cholesterol ~~does not tend to~~ progression towards the interior of the protein is significantly diminished after 100 ns (see Fig. S15 and Table S4). ~~As shown in Fig. S7, cholesterol backs away from the portal gate in nearly 50% of the trajectories (i.e. 19 out of 40 trajectories).~~ Therefore, our simulations suggest that the ability of cholesterol to access the interior of the A_{2A}R can be modulated by the nature of the membrane environment.” (Pages 6, paragraphs 2, lines 15-23)

In addition, we made a minor change to Figs S13 and S15 for the sake of consistency, namely we now show the reference distance (i.e. red horizontal lines) as the one measured from the original snapshot used to re-spawn each set of simulations (see below).

Fig S13, caption

“...(i.e. 1-10, 11-20, 21-30, 31-40). The distance at the beginning of the simulation as measured from the snapshot used to re-spawn each set of 10 trajectories ~~mean average distance for each 10 replicates (i.e. A, B, C or D) measured at the beginning of the simulation~~ is represented as one single red horizontal line in each of the graphs. Average distance for each set of replicates is reported in Table S4. Inset figures show...”

Fig. S15, caption

“...(i.e. 1-10, 11-20, 21-30, 31-40). The red horizontal line of each graph corresponds to the distance at the beginning of the simulation as measured from the snapshot used to re-spawn each set of 10 trajectories ~~mean average distance at the beginning of the simulation over all 10 replicates~~. Average distances for each set of replicates are reported in Table S4. Inset figures show...”

Table S4

Table S4 | Distance to the protein interior from short replicates

Group of replicates	Complex membrane	ICL3 included	POPC bilayer
1-10	32.78 ± 0.45	33.48 ± 0.12	38.12 ± 0.82
11-20	28.08 ± 0.48	29.43 ± 0.16	35.99 ± 0.60
21-30	20.28 ± 0.15	21.18 ± 0.10	24.76 ± 0.40
31-40	20.98 ± 0.36	19.19 ± 0.14	22.43 ± 0.36

As in Figs S13, S15, and S16, values are reported as the mean distance from the center of mass of cholesterol to protein residue E1.39 (Å) ± the standard error of the mean (SE). The last 50 ns of each trajectory were used in this analysis. The initial distance for replicates 1-10, 11-20, 21-30 and 31-40 is 32.05 Å, 31.60 Å, 21.23 Å and 19.80 Å, respectively (depicted as red horizontal lines in Figs S13, S15, and S16).

References

1. Castillo, C. A., Albasanz, J. L., Fernández, M. & Martín, M. Endogenous expression of adenosine A1, A2 and A3 receptors in rat C6 glioma cells. *Neurochem. Res.* **32**, 1056–70 (2007).
2. Liapakis, G., Simpson, M. M. & Javitch, J. a. The substituted-cysteine accessibility method (SCAM) to elucidate membrane protein structure. *Curr. Protoc. Neurosci.* **Chapter 4**, Unit 4.15 (2001).
3. Javitch, J. a, Li, X., Kaback, J. & Karlin, a. A cysteine residue in the third membrane-spanning segment of the human D2 dopamine receptor is exposed in the binding-site crevice. *Proc. Natl. Acad. Sci. U. S. A.* **91**, 10355–10359 (1994).
4. Jaakola, V.-P. & Ijzerman, A. P. The crystallographic structure of the human adenosine A2A

- receptor in a high-affinity antagonist-bound state: implications for GPCR drug screening and design. *Curr. Opin. Struct. Biol.* **20**, 401–14 (2010).
5. Piechnick, R. *et al.* Effect of channel mutations on the uptake and release of the retinal ligand in opsin. *Proc. Natl. Acad. Sci. U. S. A.* **109**, 5247–52 (2012).
 6. Park, J. H. *et al.* Opsin, a structural model for olfactory receptors? *Angew. Chemie - Int. Ed.* **52**, 11021–11024 (2013).

Reviewer#2 (R2)

Remarks to the Author:

A Summary of the key results

The key result of the article "Membrane cholesterol can access the interior of G protein-coupled receptors: the adenosine A2A receptor case" is the description of an effect of cholesterol on adenosine A2aR binding that is not purely allosteric, but seems competitive at the orthosteric site.

B Originality and interest: if not novel, please give references

This is an original observation that, potentially, could have important implications for all class of GPCRs.

C Data & methodology: validity of approach, quality of data, quality of presentation

The binding approach needs to be revised in accord to what suggested in (F)

D Appropriate use of statistics and treatment of uncertainties

Statistic is appropriate for binding data presented, except that in the Figure legends, Authors should specify how many replicates were done for each experiment, it is not enough to indicate "from, at least 3 separate experiments".

E Conclusions: robustness, validity, reliability

The robustness of the conclusions is biased by the fact that at this stage results could be interpreted alternatively, with M β CD and WSC altering the turnover of the A2aR in intact cells (see F).

F Suggested improvements: experiments, data for possible revision

This referee feels that additional binding experiments are needed to strengthen the results and exclude possible artifacts. A list of them is summarized below.

G References: appropriate credit to previous work?

References are appropriate.

H Clarity and context: lucidity of abstract/summary, appropriateness of abstract, introduction and conclusions

Abstract, introduction and conclusions are appropriate pending the exclusion of the alternative interpretation of the data (se F).

We thank Reviewer 2 for the thorough review and constructive comments on the manuscript. As we detail below, we have systematically address all comments, concerns and suggestions of the reviewer.

R2.1) *From the graph of Figure 1 it is evident that at time longer than 40 minutes M β CD decreases the binding of [3H]ZM241385. The Authors assume that the decrease in binding from 50 to 240 minutes follows the interpolation. Nevertheless, this assumption is risky; binding could fall very rapidly, as suggested by its trend between 40 and 50 minutes. This is not a detail, because in the experiment of Figure 2 cells are exposed to M β CD + WSC for 90 minutes. The Authors should clearly specify whether the compounds were used in sequence (M β CD - than washing - than WSC), or they*

were used together in the bath for 90 minutes. In the latter case, additional time points should be added to the time course of Figure 1. Exposure of the cells to M β CD for 90 minutes, could decrease the binding of [3H]ZM241385 to a level that could make data difficult to interpret.

The concentration of M β CD utilized in Fig. 1 is not reported anywhere. Is it the same utilized in Figure S1?

We thank the reviewer for pointing out inconsistencies in Figs. 1 and 2.

Regarding Fig. 1, we agree with the reviewer about the risk of interpolating data points between 50' and 240'. Accordingly, we have updated Fig. 1 by removing such interpolation. We now report this data point in the main text of the manuscript, as follows:

Results

“Longer incubation times with M β CD (i.e. 240 minutes) did not result in higher levels of A_{2A}R specific binding (469.7 ± 13.7 fmol/mg prot, n=2), likely due to a compensatory mechanism to maintain cholesterol homeostasis in treated cells.” (Page 3, paragraph 4, lines 5-10)

In addition, as suggested by the reviewer, the caption of Fig. 1 now reports the concentration of M β CD used in this experiment.

Fig. 1, caption

“Time course of 5 mM M β CD addition on [3H]ZM241385 specific binding to...”

With respect to Fig. 2, whilst in the caption of Fig. 1 we mention that M β CD and WSC are indeed added sequentially (i.e. M β CD treatment - then washing step - then WSC addition), we agree with the reviewer that this is not entirely clear. Therefore, we have improved the caption to reflect this aspect of the methodology as follows:

Fig. 2, caption

“A_{2A}R specific binding in intact cells. A_{2A}R [3H]ZM241385 radioligand binding was determined after treatment with 5 mM methyl- β -cyclodextrin (M β CD) and/or 1 mM water soluble cholesterol (WSC) ~~or both (in the specified sequential order)~~ for 40, 50 or 90 minutes. Bars 5 and 6 represent a sequential treatment (i.e. first M β CD is added, then a washing step, and finally WSC for 40 or 50 minutes). Mean \pm SEM values obtained from n=3 (columns 2, and 4 to 6), n=4 (column 3), and n=5 (column 1) separate experiments carried out in triplicate. * p<0.05, ** p<0.01 and *** p<0.001 significantly different from control value & p<0.05 and && p<0.01 significantly different from M β CD value.”

R2.2) Many membrane proteins undergoes to a continuous turnover of internalization and recycling (Hao and Maxfield, Characterization of rapid membrane internalization and recycling. *J Biol Chem.* 2000 May 19;275(20):15279-86). If internalization is inhibited the number of receptors on the membrane increase. Cholesterol is a main component of caveolae and less of clathrin coated pits, and Its depletion inhibits receptor internalization (Agathe et al., 1999 Acute cholesterol depletion inhibits clathrin-coated pit budding. *Proc Natl Acad Sci U S A.* 1999 Jun 8; 96(12): 6775-6780). A2aR are internalized through both, caveolae and clathrin coated pits and they

undergo to constitutive internalization and recycling (Klaasse et al. Internalization and desensitization of adenosine receptors. Purinergic Signal. 2008 Mar; 4(1): 21-37; Burgueño et al., The adenosine A2A receptor interacts with the actin-binding protein alpha-actinin. J Biol Chem. 2003 Sep 26;278(39):37545-52). Given that, the Authors should exclude that the effect of MβCD is due to inhibition of A2aR internalization, and redistribution of the receptors on the plasma-membrane.

We thank Reviewer 2 for bringing up such an interesting consideration on receptor internalization. We agree that further experiments are needed to demonstrate that cholesterol modulation of A_{2A}R specific binding is not mediated by receptor internalization/trafficking. To this end, we performed new binding assays in intact cells using two different inhibitors of endocytosis. As we detail below, these new experiments rule out any potential role of MβCD as an inhibitor of A_{2A}R internalization/trafficking.

As pointed out by the reviewer, cholesterol depletion has been shown to alter the structure of both clathrin-coated pits^{7,8} and cholesterol-rich caveolae⁹⁻¹¹ thus inhibiting endocytosis. On the other hand, as highlighted by Klaasse et al.¹² and Mundell & Kelly et al.¹³ in two excellent reviews on this topic, there are clearly several inconsistencies in the literature regarding both the degree and the main mechanism of A_{2A}R internalization. While some studies find the A_{2A}R to be resistant to agonist-induced internalization^{14,15}, other authors show that this receptor does undergo significant internalization in different cell models¹³.

In this context, and following the suggestions of the reviewer, we investigated the potential effect of inhibiting endocytosis on A_{2A}R specific binding by using two different cell permeable inhibitors, namely Pitstop 2 and Dynasore. These compounds have been shown to inhibit both clathrin-dependent^{16,17} and clathrin-independent^{18,19} endocytosis. In contrast to the clear increase in A_{2A}R specific binding upon MβCD treatment (Fig S8, bar 2), none of these endocytosis inhibitors altered A_{2A}R specific binding after 40 min treatment (Fig S8, bars 3 to 5) when compared to the control experiment (Fig S8, bar 1). These results address the main concern of the reviewer and demonstrate that the effect of MβCD is not due to inhibition of A_{2A}R internalization.

Thanks to the suggestion of the reviewer, we have improved the main manuscript by discussing this interesting finding:

Results

“...WSC concentrations (see Fig. S7 and section S2). **To rule out that the former effect is the result of a higher number of A_{2A}Rs available due to an inhibition of receptor internalization by MβCD, we performed new binding assays in the presence of different inhibitors of endocytosis (see details in section S1 of the SI). As shown in Fig. S8, inhibiting endocytosis does not significantly modulate A_{2A}R specific binding, hence demonstrating that receptor internalization is not involved in the cholesterol-mediated modulation of A_{2A}R specific binding.** While cholesteryl hemisuccinate, a...” (Page 4, paragraph 1, lines 3-14)

In addition, we have added a new section (i.e. section S1) to the SI describing the results from these new experiments (Fig. S8) as follows:

“Section S1 - Effect of inhibition of receptor internalization on A_{2A}R specific binding

Cholesterol depletion has been shown to alter the structure of both clathrin-coated pits¹⁸⁻²⁰ and cholesterol-rich caveolae^{21,22} thus inhibiting endocytosis. This inhibitory effect on the formation of endocytic vesicles could interfere with normal A_{2A}R trafficking (i.e. modify receptor concentration in the membrane) thus altering the outcome of specific binding assays. In order to rule out this possibility, we studied A_{2A}R specific binding in the presence of two different cell permeable inhibitors, namely Pitstop 2 and Dynasore. Dynasore is a dynamin GTPase inhibitor that prevents the scission of dynamin-dependent endocytic vesicles (i.e. clathrin- and caveolin-coated vesicles)^{23,24}. On the other hand, whilst Pitstop 2 has been commercialized as clathrin-mediated endocytosis inhibitor²⁵, it has also been shown to inhibit clathrin-independent endocytosis²⁶. As we show in Fig. S8, neither Dynasore (bars 3 and 4) nor Pitstop 2 (bar 5) was able to significantly modulate A_{2A}R specific binding after 20-40 minutes when compared to the control (bar 1). This clearly demonstrates that the increase in specific binding upon cholesterol depletion via MβCD treatment (Fig. S8, bar 2) is not due to an inhibition of receptor internalization.” (SI: Page 3)

“Figure S8 | Effect of MβCD or different endocytosis inhibitors on specific A_{2A}R binding in intact C6 glioma cells. Cells were treated for the indicated time (minutes) with MβCD or endocytosis inhibitors Pitstop 2 or Dynasore. Specific binding of 20 nM [³H]ZM241385 to A_{2A}Rs was determined as described in the Methods. These results are mean ± SEM values obtained from n=4 (columns 2, 3 and 5), n=5 (column 4) and n=6 (column 1) separate experiments carried out in duplicate. ** p<0.01 significantly different from control value.”

Methods

“...Theophylline were obtained from Sigma (Madrid, Spain). Dynasore (3-Hydroxynaphthalene-2-carboxylic acid (3,4-dihydroxybenzylidene)hydrazide) and Pitstop 2TM (N-[5-(4-bromobenzylidene)-4-oxo-4,5-dihydro-1,3-thiazol-2-yl]naphthalene-1-sulfonamide) were purchased from Abcam Biochemicals (Cambridge, UK).” (Page 11, paragraph 2, lines 7-13)

“...treatment was performed in serum-free medium. *Inhibition of endocytosis in living C6 cells.* To inhibit endocytosis, C6 cells were incubated with 80 μM Dynasore, a cell-permeable dynamin inhibitor, or 25 μM Pitstop 2TM, for 20 or 40 minutes at 37 °C (see SI for more details).” (Page 12, paragraph 1, lines 16-20)

R2.3) *If cholesterol is a competitive ligand at the orthosteric site, high concentrations of WSC should completely displace [3H]ZM241385 from the A2aR binding pocket. To address this point, a competition binding curve of WSC against [3H]ZM241385 should be done. If [3H]ZM241385 is not completely displaced, it is hard to believe that cholesterol bind to the orthosteric site, unless we assume that the orthosteric binding pocket can accommodate both cholesterol and 3H]ZM241385. Nevertheless, this can be excluded by the conclusion of the Authors "our data indicate that, if cholesterol reaches the interior of the protein, it will likely hamper the binding of A2AR ligands such as ZM241385 to the orthosteric binding site." Furthermore, a competition binding curve will consent to measure the affinity of cholesterol for the orthosteric site. Such an experiment should be done on membrane preparations, in order to avoid the confounding effect of other mechanisms that could be induced by WSC treatment in intact cells (receptor internalization for instance).*

Again, we thank Reviewer 2 for raising this interesting point. Both the results from the new experiments suggested by the reviewer and the subsequent discussion have significantly improved the manuscript.

As recommended by the reviewer, we performed competition binding experiments in membrane preparations to probe whether WSC was able to completely displace [3H]ZM241385 specific binding in a concentration-dependent manner. The results from these experiments show that high concentrations of WSC can completely abolish [3H]ZM241385 specific binding. Specifically, WSC was able to displace 20 nM [3H]ZM241385 binding with an equilibrium dissociation constant (K_i) of 233 μ M (95% Confidence Intervals 194 to 279 μ M).

It is important to note that whilst the results from biotinylation and competition binding experiments further support a competitive role of cholesterol at the orthosteric site of the $A_{2A}R$, they do not exclude a non-competitive (i.e. allosteric) effect of cholesterol. In fact our study shows that both types of modulation are likely exerted together. We believe this is a relevant aspect of our manuscript, so we have made this point clear by adding a new section to the SI where we discuss this aspect (see below).

The results from the competitive binding experiments are now reflected in the main manuscript and the SI:

Results

“This clearly suggests that cholesterol has an inhibitory effect on [3H]ZM241385 binding to the $A_{2A}R$. This effect was confirmed in membranes from control cells by competitive binding experiments in the presence of increasing WSC concentrations (see Fig. S7 and section S2). To rule out that...” (Pages 3 and 4, paragraphs 5 and 1, line 13 and 1-3, respectively)

Discussion

“...new mechanism proposed in this manuscript. Taken all together, cholesterol is likely to modulate $A_{2A}R$ ligand binding properties through a mixed-mode of action, namely both orthosteric- and allosterically. Nevertheless, further research shall be aimed to unveil the exact contribution of each mode of cholesterol-mediated modulation at this or other GPCRs” (Page 10, paragraph 2, lines 11-18)

Methods

“...*Filipin fluorescence staining*. A cell-based Cholesterol Assay Kit from Abcam (Cambridge, UK) was performed in intact C6 cells in order to visualize and measure cholesterol in intact cells by using filipin III as a fluorescence probe of cholesterol.” (Page 11, paragraph 2, lines 74-78)

“5% CO₂ at 37°C, as previously described⁴⁹. *Plasma membrane isolation*. Cells were homogenized on ice-cold isolation buffer (50 mM Tris-HCl pH 7.4, 10 mM MgCl₂ containing protease inhibitors) and centrifuged at 4 °C for 5 min at 1000xg in a Beckman JA 21 centrifuge. The supernatant was centrifuged at 4 °C for 20 min at 27000xg and the pellet was resuspended in isolation buffer as in previous work⁴⁹.” (Page 11, paragraph 2, lines 21-28)

“...were reserved for protein concentration measurement. Radioligand binding assays were also performed in plasma membranes (50-100 μ g) pre-incubated with 5 U/ml ADA for 30 minutes at 25 °C to remove endogenous adenosine using 20 or 40 nM [3H]ZM241385 for 2 h at 25 °C. Non-specific binding was obtained in the presence of 5

mM theophylline, as in previous work⁵⁰. For competition binding experiments, different (1 μ M - 3 mM) concentrations of WSC were used to displace total binding of 20 nM [³H]ZM241385. Binding was stopped by rapid filtration through Whatman GF/B filters, which were immediately washed and counted in a Microbeta Trilux liquid scintillation counter (Wallac). *Cholesterol analysis in intact cells*. Cholesterol content in intact C6 cells...” (Page 11, paragraph 2, lines 45-58)

SI

Fig. S7

“**Figure S7 | WSC competition binding curve in C6 plasma membranes.** Plasma membranes isolated from control cells were incubated with 20 nM [³H]ZM241385 and different WSC concentrations (1 μ M to 3 mM) as described in the Methods section. These results are mean \pm SEM values obtained from 3 different samples analyzed in duplicate.”

Section S2

“**Both orthosteric and allosteric effect contribute to the cholesterol-mediated modulation of A_{2A}R specific binding**

As previously postulated for other GPCRs, cholesterol is thought to modulate ligand binding from the exterior of the protein via specific binding at the surface of the receptor (i.e. direct allosteric effect) or by changing key membrane properties like fluidity or membrane thickness (i.e. indirect allosteric effect) (reviewed in refs. 27 and 28). In this line, our molecular dynamics simulations corroborate the presence of several interacting sites at the surface of the A_{2A}R (Fig. S12), as previously suggested by Lee & Lyman²⁹. Likewise, new saturation binding experiments (Fig. S5) show traits that point towards the ability of cholesterol to allosterically modulate ligand binding (e.g. major increase in B_{max} and slight increase in K_d upon cholesterol depletion). On the other hand, competitive binding experiments in the presence of increasing WSC concentrations (1 μ M to 3 mM) (Fig. S7) suggest an orthosteric mode of action of cholesterol. In these experiments, WSC is able to displace 20 nM [³H]ZM241385 binding with an equilibrium dissociation constant (K_i) of 233 μ M (95% confidence intervals 194 to 279 μ M), confirming the ability of cholesterol to compete for binding to the orthosteric site.

Nevertheless, classical binding experiments are probably not able to separate orthosteric from allosteric contribution when both modulations are exerted together. In addition, cholesterol is generally considered to bind the surface of membrane proteins (or at most at shallow sites) and hence deeply buried sites are simply overlooked, as shown by Brannigan et al. for the nicotinic acetylcholine receptor³⁰. The same assumption has likely precluded the observation of cholesterol accessing the interior of the protein in some of the previously reported cholesterol-mediated modulation of GPCRs. Overall, while the present study represents, to our knowledge, the first report of a competitive effect of cholesterol at the orthosteric site of a GPCR, future investigations shall aim to unveil the exact contribution of each type of cholesterol modulation (i.e. orthosteric versus allosteric) at the A_{2A}R or other receptors of the family.” (SI: Page 3)

R2.4) *Figure S3 shows an inhibitory effect of WSC on filipin III staining. It is important to have a control with cells that do not express A_{2A}R to confirm that the effect of filipin III is specific for cells that express the A_{2A}R.*

Based on the reviewer’s comment, we have noticed that the description of this assay is more ambiguous than intended. Filipin III is a highly fluorescent molecule that binds to cholesterol in plasma membranes, and hence specifically suited to detect fluorescent

filipin-sterol complexes. In this work, we only used filipin III to monitor the depletion (Fig. S9) or replenishment (Fig. S10) of membrane cholesterol, processes that are not influenced by A_{2A}R expression.

For clarification purposes, we have improved the caption of Figs. S9 and S10 as follows:

“Figure S9 | Cholesterol depletion from membranes monitored by filipin III staining. Phase contrast and filipin III fluorescence images of intact C6 glioma cells (A) and quantitation of filipin III fluorescence intensity (B) after ~~indicated time of treatment~~ with 5 mM MβCD ~~treatment~~ during the indicated time. ~~(A)~~ Since filipin III specifically binds cholesterol, ~~Q~~ quantitation of filipin III fluorescence intensity ~~(B)~~ monitors ~~here indicated a membrane depletion of~~ cholesterol ~~level~~ depletion. These results are mean ± SEM values obtained from ~~at least 3~~ n=5 separate experiments carried out in different cell culture passages. * p<0.05, ** p<0.01 and *** p<0.001 significantly different from control value (t:0).”

“Figure S10 | Cholesterol replenishment to intact cell membranes monitored by filipin III staining. Phase contrast and filipin III fluorescence images of intact C6 glioma cells (A), and quantitation of filipin III fluorescence intensity (B) after ~~indicated time of treatment~~ with 1 mM water soluble cholesterol (WSC) ~~treatment~~ during the indicated time. ~~(A)~~ Since filipin III specifically binds cholesterol, an adequate incorporation of cholesterol into the membrane is followed by an increase in filipin III fluorescence intensity ~~(A)~~ Quantitation of filipin III fluorescence intensity ~~(B)~~ indicated ~~an increased cholesterol level~~. These results are mean ± SEM values obtained from 3 separate experiments carried out in different cell culture passages. ** p<0.01 significantly different from control value.”

R2.5) Appropriate use of statistics and treatment of uncertainties (from point D)

Statistic is appropriate for binding data presented, except that in the Figure legends, Authors should specify how many replicates were done for each experiment, it is not enough to indicate "from, at least 3 separate experiments".

As suggested by the reviewer, we now indicate, when appropriate, the number (n) of replicates for each experiment.

Fig. 1

“...SEM values obtained from ~~at least~~ n=3 separate experiments carried out in...significantly different from control value (time 0, n=5).”

Fig. 2

“...SEM values obtained ~~from at least~~ n=3 (columns 2, and 4 to 6), n=4 (column 3), and n=5 (column 1) separate experiments...”

Fig. 6

“...in (A). Bar 1 (control, n=11): ZMA binding...effect. Bar 2 (n=5): ZMA binding...pocket. Bar 3 (n=5): biotinylation of...reduced. Bar 4 (n=4): Low biotinylation...binding. Bar 5 (n=4): lowest ZMA binding...”

Fig. S1

“...SEM values obtained from ~~at least~~ n=3 (columns 1, 2 and 4) and n=4 (column 3) separate experiments carried out in triplicate.”

Fig. S9

“...SEM values obtained from n=35 separate experiments carried out in different cell culture passages.”

References

7. Subtil, A. *et al.* Acute cholesterol depletion inhibits clathrin-coated pit budding. *Proc. Natl. Acad. Sci. U. S. A.* **96**, 6775–80 (1999).
8. Rodal, S. K. *et al.* Extraction of Cholesterol with Methyl- β -Cyclodextrin Perturbs Formation of Clathrin-coated Endocytic Vesicles. *Mol. Biol. Cell* **10**, 961–974 (1999).
9. Parton, R. G. & del Pozo, M. a. Caveolae as plasma membrane sensors, protectors and organizers. *Nat. Rev. Mol. Cell Biol.* **14**, 98–112 (2013).
10. Lingwood, D. & Simons, K. Lipid rafts as a membrane-organizing principle. *Science* **327**, 46–50 (2010).
11. Guo, S. *et al.* Selectivity of commonly used inhibitors of clathrin-mediated and caveolae-dependent endocytosis of G protein-coupled receptors. *Biochim. Biophys. Acta - Biomembr.* **1848**, 2101–2110 (2015).
12. Klaasse, E. C., IJzerman, A. P., de Grip, W. J. & Beukers, M. W. Internalization and desensitization of adenosine receptors. *Purinergic Signal.* **4**, 21–37 (2008).
13. Mundell, S. & Kelly, E. Adenosine receptor desensitization and trafficking. *Biochim. Biophys. Acta* **1808**, 1319–28 (2011).
14. Zezula, J. & Freissmuth, M. The A(2A)-adenosine receptor: a GPCR with unique features? *Br. J. Pharmacol.* **153 Suppl** , S184–90 (2008).
15. Charalambous, C. *et al.* Restricted collision coupling of the A2A receptor revisited: evidence for physical separation of two signaling cascades. *J. Biol. Chem.* **283**, 9276–88 (2008).
16. Von Kleist, L. *et al.* Role of the clathrin terminal domain in regulating coated pit dynamics revealed by small molecule inhibition. *Cell* **146**, 471–484 (2011).
17. Macia, E. *et al.* Dynasore, a Cell-Permeable Inhibitor of Dynamin. *Dev. Cell* **10**, 839–850 (2006).
18. Dutta, D., Williamson, C. D., Cole, N. B. & Donaldson, J. G. Pitstop 2 Is a Potent Inhibitor of Clathrin-Independent Endocytosis. *PLoS One* **7**, (2012).
19. Preta, G., Cronin, J. G. & Sheldon, I. M. Dynasore - not just a dynamin inhibitor. *Cell Commun. Signal.* **13**, 24 (2015).

Reviewer#3 (R3)

Remarks to the Author:

The manuscript “Membrane cholesterol can access the interior of G-protein coupled receptors: The adenosine A2A receptor case” makes the bold claim that cholesterol enters the orthosteric binding pocket of A2A and competes for ligand binding. While this is certainly an intriguing hypothesis, it is also a very surprising one, and therefore requires a very high standard of proof. Unfortunately, the data do not reach this standard, and I therefore do not recommend publication as a Nature Communication. Below I explain first why this is surprising, and then second why the data do not support the conclusion stated in the title.

We appreciate the effort and critical comments provided by Reviewer 3. We were also happy to hear that the reviewer finds our hypothesis intriguing. Based on her/his critical comments, we have now improved the manuscript with new data and additional references that further proof our hypothesis and support the conclusion stated in the title of this manuscript. We hope you find interesting the comprehensive point-to-point response we give below to your comments/suggestions.

R3.1) Results contradict published reports of cholesterol-dependent GPCR function. *The authors report in Fig. 1 that radioligand binding is enhanced upon cholesterol depletion with M β CD. All published data of which I am aware indicate the opposite — that the presence of cholesterol rather enhances or is even necessary for GPCR ligand binding and function. The authors cite some of this work, but do not mention that it contradicts their own conclusion.*

We support the reviewer’s assertion that several studies have shown that cholesterol exerts a positive modulation (i.e. enhances ligand binding and/or function) of different GPCRs. However, as we show below, cholesterol is known to modulate both positively and negatively the binding and/or function capabilities of GPCRs (extensively reviewed in ²⁰⁻²⁴). Thus, numerous reports²⁵⁻³⁸ in the literature demonstrate, as we do for the A_{2A}R in this work, that the presence of cholesterol can diminish ligand binding and/or function properties of different GPCRs (i.e. negative modulation).

To draw a clearer picture of this matter, we have comprehensively reviewed the literature and summarized in table **R3.1.1** the cholesterol-mediated modulation of ligand binding and/or functional outcome of GPCRs.

Table R3.1.1 | Effect of cholesterol presence on different GPCRs

Receptor	Ligand binding	Functional outcome
α_{1A} -adrenergic	D ³¹	D ³¹
β_2 -adrenergic	E ³⁹	E ⁴⁰
Adenosine A _{2A}	D * E ^{41,42} , or N ⁴³	D ⁴⁴ E ¹⁵ , or N ¹⁵
Cannabinoid CB ₁	D ^{32,33}	D ^{32,33}

Cannabinoid CB ₂	N ⁴⁵	N ⁴⁵
Chemokine CCR ₅	E ^{46,47}	E ^{46,47}
Chemokine CXR ₄	E ^{47,48}	E ^{47,48}
Cholecystokinin CCK ₁	E ^{49,50}	E ⁴⁹
Cholecystokinin CCK ₂	N ⁵¹	N ⁵¹
Dopamine D ₁	D ⁵²	D ⁵²
Galanin	E ⁵³	E ⁵³
Metabotropic glutamate	E ⁵⁴	E ⁵⁵
Muscarinic M ₂	D ^{34,35}	D ³⁵ E ³⁵ , or N ^{35,56}
Neurotensin NTS ₁	N ⁵⁷	Unknown
Neurokinin NK ₁	E ⁵⁸	E ⁵⁸
μ-opioid	E ^{59,60} , or N ⁶⁰ D ²⁸	E ^{59,60}
δ-opioid	E ²⁹ N ⁵⁹	D ²⁹ , or E ²⁹ N ⁵⁹
κ-opioid	D ³⁰	D ³⁰
Oxytocin	E ^{49,61–63}	E ⁴⁹
Rhodopsin	Unknown	D ^{36–38}
Serotonin 5HT _{1A}	E ^{64–66} D ^{25–27}	E ^{64–66} D ^{25–27}
Serotonin 5HT _{7A}	E ⁶⁷	E ⁶⁷

*** Results reported at the present manuscript**

Abbreviations: Enhanced (E), Diminished (D), and No effect (N)

This table shows that whilst a positive cholesterol-mediated modulation of GPCR ligand binding properties is more frequently found in the literature (i.e. 13 different receptors), cholesterol has been shown to negatively modulate ligand binding properties in as many as 8 different GPCRs. This summary demonstrates that cholesterol-mediated modulation of GPCR binding or function is highly idiosyncratic, even among receptor subtypes (e.g. opioid, cannabinoid or cholecystokinin). Furthermore, the mode and extent of cholesterol-mediated modulation seems to be greatly determined by experimental conditions like the type of ligand (e.g. agonist versus antagonist), the type of preparation (e.g. intact cells versus membrane preparations, cell membrane expression versus detergent reconstitution) (see point **R3.2** below) or the type of functional outcome probed. Thus, rather than contradicting published reports, our results add new and exciting data on the cholesterol-mediated modulation of GPCRs.

On the other hand, there is strong scientific consensus on the cholesterol-mediated stabilization of GPCR architecture^{57,68–72} (reviewed in 73 or 23). As we discuss below (see point **R3.4**), the cholesterol-mediated stabilization of rhodopsin architecture is thought to be one of the driving forces behind the clear inhibitory effect that the presence of cholesterol exerts on this protein.

We really thank the reviewer for bringing up this discussion. We have significantly improved the manuscript by incorporating the most important aspects of this subject to the main text, as follows:

Introduction

“...lipid composition in receptor function. Recent work shows that phospholipids can allosterically modulate the activity of GPCRs¹. In addition, ~~has demonstrated that~~ membrane cholesterol significantly modulates the stability, ligand binding properties, and function of several GPCRs ~~including the serotonin receptor, the β_2 -adrenergic receptor, rhodopsin and the cholecystokinin receptor~~ (reviewed in refs. 2-5). Specifically, the presence of cholesterol in cell membranes can either enhance⁶⁻¹⁰ (i.e. positively modulate) or decrease¹¹⁻¹⁴ (i.e. negative modulation) ligand binding and/or functional properties of different GPCRs (see Table S1 for a comprehensive summary).” (Page 2, paragraph 1, lines 1-11)

“Whether this modulation is exerted through indirect effects^{19,20} (i.e. changes in membrane properties), direct interactions²¹⁻²⁴ between cholesterol and GPCRs, or both, has for long been a matter of intense debate²⁻³ (see ref. 5 for a recent review on this topic). ~~While changes in membrane properties induced by cholesterol can indirectly alter GPCR architecture^{4,5}, specific cholesterol binding sites have also been identified at prototypical GPCRs⁶⁻⁹.~~ Specific cholesterol binding sites have been identified at the surface of different GPCRs, ~~Identification of these binding sites has~~ suggested a potential allosteric role of cholesterol in modulating GPCR function ~~through lipid-protein interactions at allosteric sites in the receptor surface~~. Intriguingly, other studies...or the chemokine receptor CXCR2²⁶. Similarly, oxysterol derivatives are known allosteric modulators of the oncoprotein Smoothed²⁸ (SMO), a class F GPCR. A very recent crystal structure of SMO shows one cholesterol molecule in the binding site of the extracellular domain of this receptor²⁹.” (Page 2, paragraph 3, lines 1-24)

In addition, please note that we have added the comprehensive revision of the literature we have made here to the SI (now Table S1).

R3.2) *For example, the authors cite O'Malley et al's work (ref 21) in which A2A is reconstituted into detergent micelles, and in which it is found that that the receptor does not bind ligand unless a sterol derivative (cholesterol hemisuccinate, in this case) is present in sufficient quantity. The present manuscript use the O'Malley result to support their simulation results, in particular that they find a similar number of cholesterol in the vicinity of the receptor. However, they fail to mention that the O'Malley data contradict their own conclusions.*

We agree with the reviewer that the work from O'Malley and co-workers⁴¹ show a clear positive dependence of cholesteryl hemisuccinate (CHS) on A_{2A}R activity. It is important to note that, as shown for other GPCRs (see point 1 below), results from detergent-reconstituted micelles are hardly comparable to those obtained using native membrane environments (e.g. intact cells) (see points 2 and 3). Nevertheless, the results from O'Malley et al.⁴¹ rather depict a well-established property of cholesterol as solubilizer agent and stabilizer of protein structure in detergent-mediated extractions of GPCRs⁶⁸ (see also points **R3.1** and **R3.4**). As we rationalize below, the results we present in this manuscript do not contradict the work from O'Malley and co-workers nor any previous study.

1. The difference raised by the reviewer has already been shown for various GPCRs. Thus, addition of cholesterol additives such as CHS to detergent micelles enhances the stability (e.g. recovers ligand binding properties) of GPCRs including the κ -opioid⁷⁴, the muscarinic M₂⁷⁵, or the neurotensin NTS₁ receptor⁷⁶. However, in native membrane environments (i.e. intact cells or cell membrane preparations), the presence of cholesterol either decreases^{30,34,35} or have no effect⁵⁷ on the ligand binding properties of the same receptor.
2. Commonly used sterol additives like CHS and cholesterol can exert different effects at the same GPCR. While CHS and cholesterol are generally thought to confer similar properties to the lipid bilayer⁷⁷, the much higher polarity of this analog likely modulate GPCRs through a different mechanism. For example, as shown by Kimura et al.⁷⁸, CHS is able to enhance G protein activation at the cannabinoid CB₂ receptor whereas the addition of cholesterol does not produce any effect.
3. Several other aspects of the methodology employed by O'Malley et al.⁴¹, or similar experiments using detergent-solubilized GPCRs, make the comparison to our results (i.e. native-like conditions) simply not meaningful.
 - a. Detergent-solubilized GPCRs do not necessarily represent a physiological state of the receptor. Like O'Malley et al.⁴¹, most studies using solubilized GPCRs do not use a native form of the receptor. Instead, GPCR purification frequently requires an artificial elongation/truncation of the protein sequence. In the case of O'Malley et al.⁴¹, the authors purified the A_{2A}R by adding a deca-histidine motif to the receptor, a common protein modification employed to improve protein purification with affinity chromatography.
 - b. Detergents employed in GPCR purification do not adequately represent the native membrane environment of the receptor. Thus, increasing concentrations of certain detergents have been shown to inhibit ligand binding⁷⁹, presumably by changing the architecture of GPCRs. As a matter of fact, as shown by O'Malley et al.⁴¹, common detergents like the n-Dodecyl β -D-maltoside (DDM) are able to irreversibly inhibit ligand binding at the A_{2A}R prior to the addition of cholesterol. While cholesterol additives basically counteract the inhibitory effect of the detergent and recover ligand binding properties, the mechanism behind such stabilization and whether the exact native fold of the receptor is preserved is still a matter of debate.

Taken as a whole, based on previous reports (point 1) and clear differences on the methodology employed by O'Malley et al.⁴¹ (points 2 and 3), our findings do not contradict their results. Instead, we provide a more physiological picture of cholesterol-mediated effect on A_{2A}R ligand binding properties and suggest a completely new mode of cholesterol-mediated modulation of GPCRs.

Nevertheless, we understand the concern raised here by the reviewer. Therefore, to avoid any possible misunderstanding, we now make clear in the manuscript that in their

work O'Malley et al.⁴¹ report a positive cholesterol-mediated modulation of A_{2A}R activity in detergent micelles, see below:

“modulation of A_{2A}R specific binding. While cholesteryl hemisuccinate, a cholesterol derivative, enhances the stability and activity of detergent-solubilized A_{2A}Rs³⁷, our data clearly shows suggests that naturally-occurring cholesterol has an inhibitory effect on [³H]ZM241385 binding to the A_{2A}R in more physiological environments (i.e. intact cells and cell membrane preparations). Our findings goes along with previous experimental evidence in several various class A GPCRs reportingpointing towards a negative cholesterol-mediated modulation of ligand binding (Table S1)^{18,19}.” (Page 4, paragraph 1, line 14-25)

R3.3) *The authors also mention other reports of cholesterol mediated GPCR function (refs 18 and 19), stating that “Our finding goes along with previous experimental evidence in various class A GPCRs pointing towards a cholesterol-mediated modulation of ligand binding.” However, they fail to mention that refs 18 and 19 also contradict their own conclusion. This is a bit disingenuous, to put it mildly.*

We regret that these statements have led reviewer 3 to misjudge our intentions. As we thoroughly described in point **R3.1**, the presence of cholesterol is clearly able to exert a negative modulation (i.e. decrease) of ligand binding at several GPCRs^{25,27,28,30–35,52}, and thus the results we report in this manuscript do not contradict previous findings. While reference 18²⁷ indeed demonstrate that cholesterol depletion enhances ligand binding at the serotonin 5HT_{1A} receptor, reference 19⁴⁶ shows that cholesterol is, in contrast, necessary for the binding and functional properties of the chemokine CCR₅. Therefore, by mentioning these two references we just aimed to exemplify the cholesterol-mediated modulation of GPCRs.

Nevertheless, to prevent future misinterpretations, and thanks to the comments from the reviewer, we have significantly improved the state-of-the-art literature of the manuscript regarding the cholesterol-mediated modulation of GPCRs (see details in points **R3.1**, **R3.2** and **R3.4**).

R3.4) *Lastly, rhodopsin clearly requires significant cholesterol to function, as the rod outer cell membrane contains ~ 40 mol% cholesterol. It is therefore incumbent upon the authors to discuss and explain these discrepancies, and to explain why their experimental protocol yields different results compared to previously published data.*

We thank the reviewer for bringing up the modulation of rhodopsin by cholesterol as this is the prototypical case of the membrane-mediated modulation of GPCRs. However, as a matter of fact, the presence of cholesterol have been widely demonstrated to reduce (i.e. negatively modulates) the function of rhodopsin.

It is true that disk membranes in rod outer segments of retinal photoreceptor cells are rich in cholesterol, to be more precise, cholesterol content in these membranes ranges from 30 mol % (i.e. newly formed basal disks) to 5 mol % (i.e. older apical disks)^{80–82}. However, to the best of our knowledge, all previous reports in the literature (reviewed in refs. ^{83–85}) show that cholesterol clearly inhibits rhodopsin function by hampering the formation of the active intermediate metarhodopsin II (MII)^{36–38}. While the mechanism behind such inhibition is still a matter of intense debate⁸⁶, different studies^{72,87}

demonstrate that cholesterol stabilization of metarhodopsin I (MI) architecture shifts the MI-MII equilibrium towards MI thus inhibiting the formation of MII.

On the other hand, molecules of similar size and amphipathic nature like detergent Octyl β -D-glucopyranoside can also stabilize rhodopsin by accessing/exiting the binding pocket⁶ from the membrane side using the same route we propose in this manuscript. It is, therefore, worth to speculate that cholesterol could exert part of its inhibitory action on rhodopsin via accessing the interior of the protein. It is worth quoting from a very recent review²³ of a renowned expert in cholesterol-mediated modulation of GPCRs, Dr. Gerald Gimpl: ‘*until now, cholesterol molecules have not been observed to bind inside the helix bundle of a GPCR. However, given that >95% of the receptor structures are unknown it is not excluded that cholesterol could also once be observed within the interior of the helix bundle of a GPCR*’.

The interesting discussion raised by the reviewer has made us improve the manuscript to better capture the relevance of the cholesterol-mediated modulation of rhodopsin to our findings (see below)

Introduction

“...A well-known example of this modulation is observed in rod outer segments for the prototypical receptor rhodopsin, where higher cholesterol concentrations in newly formed basal disks are used by these cells to stabilize the structure of metarhodopsin I (MI), thus hampering the formation of the active intermediate metarhodopsin II (MII)¹⁵⁻¹⁸.” (Page 2, paragraph 2, lines 11-19)

“...into the receptor transmembrane bundle. Different lipophilic ligands that bind to the orthosteric site of class A GPCRs are suggested to access the protein from the membrane milieu (reviewed in 5). Interestingly, in a recent crystal structure of rhodopsin³⁶, a molecule of a commonly used detergent (i.e. n-octyl β -D-glucopyranoside) was shown to stabilize the receptor by replacing retinal from the ligand-binding pocket. Therefore, as recently discussed by Gimpl⁵, it would seem plausible that cholesterol can access the interior of class A GPCRs like the A_{2A}R. To validate this new mechanism of action...” (Page 2, paragraph 5, lines 1-12)

Discussion

“...to the retinal gateway present in the opsin receptor. Since molecules of a similar size and amphipathic nature have been shown to access the retinal binding pocket³⁶, it is reasonable to speculate that the negative modulation exerted by cholesterol on rhodopsin function¹⁵⁻¹⁸ could be partly mediated by the new mechanism proposed in this manuscript. Taken all together, cholesterol...” (Page 10, paragraph 2, lines 3-9)

R3.5) *One potential source of discrepancy may be in the interpretation of the cholesterol depletion experiments. Cholesterol depletion and cholesterol content assay The authors state that treatment of the cells with 5 mM M β CD for 30 min depletes “most of the cholesterol from the cell membranes.” Though the authors are not explicit about what is meant by “most”, one assumes that it must mean at least 51%. (Note that visual inspection of Fig. 1 is not sufficient to answer this question, as the data indicate some unexplained variability of cholesterol content with time of M β CD treatment. Why does cholesterol content increase from 30 to 40 min of M β CD treatment?)*

First of all, we have slightly improved Fig. 1 for better readability (e.g. higher number of ticks in the right y axis). As shown by this figure, treatment with 5 mM M β CD for 40

min depletes, specifically, 69% of the cholesterol from the membrane. We acknowledge the variability observed within 30-40 min M β CD treatment and believe this could just reflect the accuracy of the kit we employed to measure cholesterol in intact cells. Please note that regardless of this variability, the main message of Fig. 1 remains the same, namely that we report a clear inverse relationship between cholesterol levels and A_{2A}R specific binding.

Nonetheless, thanks to the suggestion made by the reviewer in point **R3.7**, we now report the exact membrane cholesterol content upon 40 min M β CD treatment, namely 61%, as obtained from lipidomics analysis (see Fig. S2).

R3.6) *This level of depletion is not supported by reports in the literature, for example Mahammad and Parmryd, BBA 1778:1251(2008), which carefully assayed cholesterol removal from membranes by M β CD, and found at most 35% reduction in membrane cholesterol levels. (Note that these authors also report that M β CD is promiscuous — it removes lipids other than cholesterol from the membrane.)*

We agree that Mahammad & Parmryd⁸⁸ carefully assayed cholesterol depletion by M β CD in T cells. However, the 35% reduction in membrane cholesterol highlighted by the reviewer was only achieved when experiments were carried out at 0°C. The latter experiments only confirmed that cholesterol removal by M β CD is much less efficient at low temperatures. As a matter of fact, as these authors report in Fig 1 of this study⁸⁸ (i.e. left picture of Fig. **R3.6.1** below), 3mM M β CD treatment for 15-20 min is enough to deplete more than 50% of the membrane cholesterol when assays are carried out at 37 °C. Please note that this is the same temperature we used in our experiments. Moreover, in these experiments the level of cholesterol depletion shows a clear increase with M β CD concentration (see left plot in Fig. **R3.6.1**). As we show in the right picture of Fig. **R3.6.1** below, treating cells with 5mM M β CD (i.e. the concentration used in our experiments) would potentially result in 80-85 % cholesterol depletion.

Fig. 1. Cholesterol extraction from Jurkat T cells by M β CD at 37 °C. 10×10^6 Jurkat T cells/ml labelled with [³H]-cholesterol were treated with M β CD in serum-free RPMI medium at 37 °C. At the indicated times, the cells were pelleted by a brief centrifugation and the M β CD-containing supernatant was transferred to a fresh tube. The cell pellet was resuspended in PBS and aliquots from both fractions were subjected to scintillation counting. Data shown are means \pm SD, n = 3.

Figure R3.6.1 | Cholesterol depletion at 37°C as reported by Mahammad & Parmryd⁸⁸. Left picture shows Fig. 1 of the study used as an example by the reviewer, where these authors show different levels of cholesterol depletion at 37 °C using different M β CD concentrations. The right picture uses the data shown in the left picture to plot the linear relationship between cholesterol levels and M β CD concentration.

Moreover, while levels of M β CD cholesterol depletion of up to 100% have been previously described in the literature⁸⁹, these values can be highly variable based on the methodology/cell type, and range from 30 to 100% (see Table 1 from an excellent review on this topic by Zidovetzki & Levitan⁸⁹). Regarding M β CD promiscuity, we agree that M β CD ability to remove other lipids has been previously suggested. However, it has also been suggested that this side effect is not significant due to the much higher specificity of cholesterol versus other membrane lipids^{88,89}. It is also worth noting that, albeit the former aspect is still under debate, incubation with M β CD is currently the most common technique used to deplete cholesterol from cell membranes⁸⁹.

All in all, both Mahammad & Parmryd⁸⁸ and previous literature reports do support the level of cholesterol depletion we show in the present study. Please note that lipidomics analysis (i.e. as suggested by the reviewer) further supports the former statement and dissipates all concerns and doubts in this regard (see point **R3.7** below).

R3.7) *Cholesterol is then reintroduced after depletion using water soluble cholesterol, and measured with a different assay (Filipin staining), on the basis of which it is stated that cholesterol has recovered to some extent. These data are presented in the SI, and are not convincing, as they are not a direct measure of cholesterol content and are not easily compared to the initial assay. Overall, the data on cholesterol content after depletion and following reintroduction using water soluble cholesterol need to be corroborated, e.g., by lipidomic analysis.*

We thank the reviewer for this interesting suggestion. We agree that further experiments such as lipidomics analysis will reliably provide data on cholesterol content after M β CD or water soluble cholesterol (WSC) treatment. Thus, we performed targeted lipidomics analysis of cholesterol content in membrane preparations isolated from control, 5 mM M β CD (40 min) and 1 mM WSC (50 min) treated cells. The results from the lipidomics analysis experiments (see Fig. S2) confirm that M β CD depletes up to 61% of cholesterol from cell membranes, whereas WSC treatment produces a 7-fold increase in membrane cholesterol levels. As we now show in Fig. S6, further radioligand binding assays using the former membrane preparations confirm the effect we describe in Figs. 1, 2, and S1 using intact cells.

Along with 2 new figures showing these results (Figs. S2 and S6), we now describe this interesting validation in the main text of the manuscript, as follows:

Results

“Our data indicate that M β CD is able to deplete ~~around most of the cholesterol~~ 70-80% of membrane cholesterol ~~from cell membranes~~ after 30 min. To accurately assess the level of cholesterol depletion, we carried out targeted lipidomics in plasma membranes. As shown in Fig. S2, 40 min treatment with 5mM M β CD depletes up to 61% of cholesterol from the membrane. In addition, further radioligand binding assays using the former membrane preparations (Fig. S6) confirm the effect we describe in Figs. 1, 2, and S1 using intact cells” (Page 3, paragraph 2, line 12-22)

“Adequate cholesterol ~~depletion and~~ insertion into the membrane was monitored in intact cells and plasma membrane fractions by filipin fluorescence staining (Figs. S9 and S10) ~~in intact cells and by~~ targeted lipidomic analysis of plasma membrane fraction (Fig. S2). Interestingly, addition of cholesterol...” (Page 3, paragraph 5, lines 1-6)

Figures

Figure S2 | Cholesterol quantification in C6 plasma membranes. Cholesterol (Ch) concentration was measured by targeted lipidomics in plasma membranes isolated from control (n=3), 5 mM M β CD (40 minutes, n=2) and 1 mM WSC (50 minutes, n=2) treated cells as described in the Methods. Data are mean \pm SEM values from indicated n samples. ** p<0.01 significantly different from control value.

Figure S6 | Effect of M β CD and WSC on specific A_{2A}R binding in C6 plasma membranes. Plasma membranes were isolated from control (n=3), 5 mM M β CD (40 minutes, n=3) and 1 mM WSC (50 minutes, n=2) treated cells. Specific [³H]ZM241385 binding (i.e. 40 nM) to A_{2A}R was measured as detailed in the Methods. These results are mean \pm SEM values obtained from the indicated number samples (n) analyzed in triplicate. * p<0.05 and ** p<0.01 significantly different from control value.

Methods

“...bovine serum albumin as standard. *Lipidomic analysis.* A complete description of the methodology employed for lipidomic analysis is shown in the methods section of the SI” (Page 12, paragraph 1, lines 34-36)

Methods SI

“**Lipidomic Analysis.** Plasma membrane preparations were isolated from control (n=3), 5 mM M β CD 40’ (n=2) and 1 mM WSC 50’ (n=2) treated cells. Methanol (0.1 ml) were added to 0.1 ml of membrane samples (containing deuterated cholesterol D7 as internal standard) and vortexed for 10s. Then, 0.2 ml of chloroform were added and vortexed for 10s. The samples were centrifuged at 1000g during 15 min at 4°C and chloroform phase (lower) were separated in a glass tube. This last step was repeated two times more. Finally, chloroform phase was evaporated using a Speed Vac (Thermo Fisher Scientific, Barcelona, Spain) and resuspended in 50 μ l of methanol/ chloroform (3:1)^{1,2}. Lipid extracts were subjected to mass-spectrometry using a HPLC 1290 series coupled to an ESI-Q-TOF MS/MS 6520 (Agilent Technologies, Barcelona, Spain). LC/MS analysis was performed following a previously described method³. In short, 2 μ L of lipid extract were injected onto an XBridge BEH C18 shield column (100 mm \times 2.1mm ID \times 1.7 μ m; Waters, Milford, MA, USA) kept at 80°C. Mobile phases, delivered at 0.5 mL/min, consisted of ammonium formate (20 mM at pH 5) (A) and methanol (B). The gradient started at 50 % B and reached 70 % B in 14 min. This was followed by a slow gradient of 70–90 % B in 50 min and an isocratic separation of 90 % B during a 15-min period. Mobile phase B subsequently reached 100 % in 5 min, and was maintained so for an additional 5 min. This method allows the orthogonal characterization (based on exact mass (<10 ppm) and on retention time) of lipids. This strategy is useful for attributing potential identities with low uncertainty when combined with internal standards. Data were collected in both positive and negative electrospray ionization TOF mode operated in full-scan mode at 100-3000 m/z in an extended dynamic range (2 GHz), using N₂ as nebulizer gas (5 L/min, 300 °C). The capillary voltage was 3500 V with a scan rate of 1 scan/s. MassHunter Data Analysis Software (Agilent Technologies, Barcelona, Spain) was employed to record results and MassHunter Qualitative Analysis Software (Agilent Technologies, Barcelona, Spain) was used to obtain the molecular features of the samples, as previously described⁴.” (SI: page 1)

R3.8) *Lastly, after 30 min of M β CD treatment, the cells are at the least very stressed, and most likely rapidly dying. Thus other mechanisms may well be responsible for the observed reduction in ligand binding upon M β CD treatment. Perhaps less protein is available at the surface, for example?*

The reviewer raised a very interesting point here. We agree that rapid and significant loss of membrane cholesterol could indirectly induce a decrease in cell viability. To rule out the possibility that either M β CD or WSC is leading to any lethal effect on cells, we studied cell viability using the XTT method⁹⁰ at 20, 40 and 60 minutes of M β CD or WSC treatment. As shown in Fig. S3A, neither M β CD nor WSC treatments affected cell viability. In fact, a visual inspection of cells up to 100 minutes after treatment confirmed no significant effects on cell morphology, division processes or number of cells (see Movie S1 and Fig. S4). Moreover, as displayed in Fig. S3B, protein content did not significantly change after 20, 40 and 60 minutes incubation with either M β CD or WSC.

Thanks to the comments made here by the reviewer, we have increased the robustness of the results we present here by excluding any loss of cell viability potentially induced by M β CD or WSC treatments. We have updated the manuscript accordingly, as we show below:

Results

“...To rule out any cytotoxic effect of M β CD or WSC treatment, cell viability was determined at 20, 40 and 60 minutes after M β CD or WSC treatments using the XTT method (see Methods). As shown in Fig. S3A, neither M β CD nor WSC treatments affected cell viability. Likewise, protein content did not significantly change after 20, 40 and 60 minutes incubation with either M β CD or WSC (Fig. S3B). Moreover, cells did not display any significant change in number (Fig. S4), morphology (Movie S1), or division processes.

Remarkably, the lack of cholesterol increases...” (Page 3, paragraph 3)

Figures

“**Figure S3 | Effect of M β CD or WSC treatment on C6 cells viability and protein level.** Cell viability (A) and total protein amount (B) were measured in C6 intact cells at indicated time of treatment and compared to control cells values. These results are mean \pm SEM values obtained from 4 independent assays carried out in sextuplicate (A) or duplicate (B).”

“**Figure S4 | Effect of M β CD or WSC treatment on C6 cells growth.** Cells were counted from phase contrast images recorded (1 frame/minute) for 100 minutes at the indicated interval and relative to the number of cells at the beginning of treatment. These results are mean \pm SEM values obtained from 2 independent assays performed at different cell passage.”

Methods

“*Cell viability assay.* Cells were seeded (10⁴ cells/well) and grown in 96-well tissue culture plate and incubated with 0.3 mg/ml XTT solution (sodium 3'-[1-(phenylaminocarbonyl)-3,4-tetrazolium]-bis (4-methoxy-6-nitro) benzene sulfonic acid hydrate) for 30 minutes at 37 °C in control, M β CD or WSC treated cells. The cleavage of XTT to form an orange formazan dye by viable cells was monitored by reading absorbance at 475 nm and 690 nm according to the manufacturer's protocol (Cell

Proliferation Kit II, Roche, Mannheim, Germany). *Protein determination*. Protein...” (Page 12, paragraph 1, lines 20-31)

R3.9) *Molecular simulation results: the structural explanation for the biochemical results is obtained on the basis of molecular simulations, but there are also some significant questions regarding the simulation protocol and therefore the conclusions. Intracellular loop 3 (between helix 6 and 7) is not included in the simulation. According to the SI at the top of the first column on page 11, the authors indicate that this loop is not included in the simulation, as it was not resolved in the crystal structure. Given that the conformation of helix 6 must undergo significant displacement to allow the cholesterol to enter (it enters between helices 5 and 6), this is a significant omission. Why did the authors not perform additional simulations that include the loop? Given the simulation time devoted to the present results, it would be a simple matter to build in the loop (perhaps from one of the Tate lab/Heptares structures) and relax it with a few microseconds of unbiased MD.*

The reviewer raised here an interesting consideration. We agree that new simulations would help to rule out any effect potentially exerted by the presence of the intracellular loop 3 (ICL3) of the A_{2A}R. Thus, we performed a new batch of simulations where we included the ICL3 taken from one of the Tate lab/Heptares structures, namely PDB:3PWH.

As shown in Fig. S16 and Table S4, the inclusion of the ICL3 loop does not have any significant influence on the results we show in this manuscript.

The results of the new simulations are now included in Table S4 and Fig. S16 of the SI:

Methods

“To rule out an effect of ICL3 omission, we performed a similar set of simulations where the ICL3 was included (see SI methods and Fig. S16).” (Page 13, paragraph 1, lines 1-3)

SI methods

“In simulation set 4 (i.e. ICL3 included), after 10000 steps of minimization, a gradual release of different applied harmonic constraints was used during 40 ns phase in the NPT ensemble. Harmonic restraints were first applied during 10 ns to all atoms of the system except for the intracellular polar head region of all membrane lipids. Constraints were then released for all water and ion atoms and the system was simulated for another 10 ns. Subsequently, the ICL3 region was released and the system (i.e. only protein and target cholesterol restrained) was further equilibrated for 20 ns.” (SI: page 2, paragraph 2, lines 3-9)

“*Inclusion of ICL3.* The ICL3 structure was taken from PDB:3PWH and included in the A_{2A}R structure using MODELLER^{v9.10}¹⁶ and VMD¹². The system was first minimized and the ICL3 subsequently equilibrated and relaxed for 50 ns in the NPT ensemble (see simulation protocol above).” (SI: page 2, paragraph 4)

SI

“**Figure S16 | Short simulation replicas of cholesterol entrance (ICL3 included).** Short simulations of the A_{2A}R (ICL3 included, as described in the methods and SI methods) and target cholesterol embedded into the original bilayer (see Tables S2 and S3). As in Figs. S13 and S15, boxplots display the distance between the center of mass of cholesterol and residue E1.39 for a set of 40 replicate simulations of 100 ns. 4 different starting positions (A, B, C and D) re-spawned from the original cholesterol

entrance trajectory were used to run each 10 replicates (i.e. 1-10, 11-20, 21-30, 31-40). The red horizontal line of each graph corresponds to the distance at the beginning of the simulation as measured from the snapshot used to re-spawn each set of 10 trajectories. Average distance for each set of replicates is reported in Table S4. Inset figures show the initial structure of the A_{2A}R (in blue) and cholesterol residue (in orange) used to start each set of simulations. E1.39 residue displayed as van der Waals spheres. The BENDIX⁸³ plugin for VMD was used to depict protein helices. Protein loops were omitted for clarity.”

R3.10) *The cholesterol entrance event was observed in 1 out of 4 simulations. Configurations were then taken from this one trajectory and used to seed other simulations, ostensibly as some sort of statistical test. However, this does not provide any information on whether the results are reproducible, since only the “interesting” event is considered in the follow up.*

We agree that the follow-up simulations we performed (i.e. 3 systems x 40 replicas x 100 ns) cannot be used to study complex aspects like what the energetic cost of cholesterol accessing the interior of the A_{2A}R or how rare this event is. However, we would like to highlight that rather than trying to address the former aspects, these simulations had two main objectives, as follows:

1. First, to explore the cholesterol entry pathway by providing more details on the tendency of this cholesterol molecule to progress towards the interior at different stages of the invasion. For example, these simulations were used to better understand the exact sequence of events as depicted in Fig. 4, namely from cholesterol binding at a specific area of the receptor surface to the subsequent cholesterol invasion of the receptor interior.
2. Second, to rule out that the process of cholesterol invading the interior of the protein was not the result of a simulation artifact observed in one single trajectory. While our simulations are not suitable to shed light on the energetic of such event, they do provide enough statistics to rule out a simulation artifact.

On the other hand, as we discuss in the next point (**R3.11**), we are now further exploring the energetic landscape of cholesterol entrance/exit routes by means of different biased molecular dynamics techniques.

We believe that we failed to adequately draw the main objectives of the follow-up simulations performed in this paper, and thus we have now slightly improved the main text, please see below:

Results

“orthosteric binding pocket of a GPCR.

To exclude simulation artifacts and better explore the cholesterol entry pathway ~~provide statistical support to this finding~~, we performed new simulations and...” (Page 5, paragraph 3, lines 1-3)

R3.11) *The authors also perform umbrella sampling in an effort to determine the cost of extracting cholesterol from the binding pocket via the aqueous phase. Why did they not perform this calculation for the entrance/exit of cholesterol via the membrane?*

Wouldn't this be an obvious way to corroborate the single event observed in the unbiased simulation?

We agree this is an interesting line of study. Since we demonstrate by different experimental techniques that cholesterol can access the interior of the A_{2A}R, the calculations suggested here by the reviewer are, however, out of the scope of the present manuscript (see point A below). In addition, as we explain in points B and C below, obtaining converged free energy values of such event is certainly not a trivial task. Nevertheless we are currently working on a follow-up study (see point B below) to fully characterize the energetics of cholesterol accessing/exiting the A_{2A}R via the transmembrane bundle. Very interestingly, preliminary results of this follow-up study (see point B below) show that cholesterol tends to access the A_{2A}R through helices TM5-TM6, in agreement with the results reported in the present manuscript.

- A) Present manuscript: as we thoroughly explain in Section S4 of the SI, the only aim of the umbrella sampling simulations reported in this manuscript was to study the cost of cholesterol exiting the receptor via the aqueous phase. These simulations demonstrate that the cost of cholesterol exiting via the extracellular water phase is very high in comparison to the transmembrane bundle. In addition, we suggest TM1-TM7 and/or TM1-TM2 openings as two feasible cholesterol exit pathways. However, the suggested exit routes are only accessory to the main objective of the umbrella sampling simulations, namely, studying the energetic cost of cholesterol egress through the water phase.
- B) Follow-up study: while exploring the energetics of spontaneous entry/exit routes by all-atom molecular dynamics (MD) simulations is outside the scope of the present manuscript, this is a very exciting line of research that we are currently addressing in a follow-up study. In fact, preliminary results from this new study using well-tempered metadynamics simulations⁹¹ confirm the tendency of cholesterol to egress the A_{2A}R binding pocket through helices TM1-TM7 or TM1-TM2. Strikingly, these simulations show that after a comprehensive exploration of the A_{2A}R surface, cholesterol happens to re-gain access to the interior of the protein through the same portal gate we suggest in this manuscript (i.e. TM5-TM6). An example of these preliminary simulations has been made available to the reviewer (see video **R3.11.1**). However, while we believe the former results are quite promising, these ongoing simulations are far from full convergence (see point B below) and hence no free energy values can be extracted at the present time.
- C) Technical challenges: previous attempts to study the egress pathway of ligands out of the binding pocket of GPCRs have already shown the complexity of this type of work using enhanced sampling MD techniques. In particular, two separate studies solely devoted to this topic suggested potential ligand exit pathways for rhodopsin⁹² and the β_2 -adrenergic receptor⁹³ only by means of random accelerated MD. It is worth noting that no free energy calculation of any of the suggested pathways was computed at this stage. Recent new approaches to enhance sampling including Gaussian accelerated MD^{94,95} and metadynamics⁹⁶ have been used to obtain ligand binding free energies. In the particular case of our simulations, the main challenge also lies in the impressive amount of calculation time needed to reach adequate convergence. Cholesterol exploring the interior of the protein prior to egress the receptor can be likely sampled in long but reasonable time scales. In contrast, higher

time scales will be needed to adequately sample the event of cholesterol binding the target surface of the A_{2A}R prior to receptor invasion. This is due to the high time scales required to account for membrane lipids unbinding from the receptor surface and overall lipid diffusion in such a complex membrane.

R3.12) *At the top of the second column on page 5, the authors speculate about the role of the membrane environment in driving cholesterol into the receptor. The simulation in which cholesterol enters the receptor includes a rather complex membrane environment, with 5 different lipids plus cholesterol, including PUFA and sphingolipid. Based on a comparison to a simulation in chol/POPC bilayer, they conclude that “an unsaturated lipid along with a cluster of 4 cholesterol molecules seem to assist cholesterol entering the receptor...” Needless to say, these are anecdotal at best, and are not sufficient to support the statement (at the end of the paragraph) that the membrane environment assists in the cholesterol entry.*

We acknowledge that such statement is rather anecdotal. Please note that we never used this comment in the manuscript to support any finding. Instead, we have used it to introduce the aspect of membrane-mediated effect on cholesterol entry that we subsequently address using further simulations (Fig. S15 and Table S4).

We have now slightly modified the main manuscript to avoid misunderstandings.

Main text

“The nature of the membrane environment ~~seems to~~ **could** be one of the driving forces behind the spontaneous cholesterol entrance into the A_{2A}R. As shown in Movie **S32**, an unsaturated phospholipid (in yellow) along with a cluster of 4 cholesterol molecules seem to **influence assist** cholesterol **entrance entering the receptor** by preventing it from diffusing back to the membrane bulk. To study the impact of membrane composition, we...” (Page 6, paragraph 2, lines 1-9)

Movie S3 caption

“...surrounding membrane bulk **seem to** assist the entrance of cholesterol (red) into the A_{2A}R (purple) **by preventing it from diffusing back to the membrane bulk**. All lipids shown ~~correspond~~ **belong** to ...”

R3.13) *I am also not certain at all whether the complex membrane is “thicker and more compact” than the POPC/chol control. The complex membrane has saturated chains and cholesterol, but it also has a significant amount of PUFA.*

We agree with the reviewer that due to complexity of this membrane, one cannot intuitively assume a certain membrane thickness. Therefore, we computed the average membrane thickness across both sets of simulations (i.e. complex membrane and POPC membrane). As shown in Fig. S14, complex membranes are clearly much thicker (i.e. around 8 Å thicker) than POPC membranes. Specifically, our simulations show average membrane thickness (i.e. distance between phosphorous atoms) of 47.80 Å and 40.08 Å for complex and POPC membranes, respectively.

Thanks to the comment from the reviewer, we now mention this fact in the main text and have added this calculation to the SI, as follows:

Main text

“...in the absence of a more compact and thicker membrane (see Fig. S14), cholesterol does not...” (Page 6, paragraph 2, line 16)

SI

“Figure S14 | Membrane thickness of complex and POPC membranes. Average membrane thickness (Å) (i.e. distance between phosphorous atoms) was computed using the MEMBPLUGIN⁸⁴ of VMD. All trajectories analyzed in this manuscript (see Table S2) were used to obtain average values.”

Extra changes to the manuscript

We detail here all extra improvements we have made to the text during the review process. These changes basically add extra information missing in the previous version, correction of typos and mistakes, etc.

Introduction

“...MD simulation and experimental results **suggest show**, for the first time **to our knowledge**, that cholesterol can compete with orthosteric ligands...”(Page 3, paragraph 1, lines 3-6)

Methods

“**Liquid scintillation solutions were purchased from Perkin Elmer (Boston, MA, USA).** All other products were of analytical grade...” (Page 11, paragraph 2, lines 13-14)

“cholesterol (WSC) (see Fig. 2 ~~and Fig. S2~~). Adequate cholesterol **depletion and** insertion into the membrane was monitored by filipin fluorescence staining (Figs. **S9 and S10**) and...” (Page 3, paragraph 5, lines 1-4)

“After incubation, cells were washed with ice-cold buffer **and disrupted** with 0.2% SDS” (Page 11, paragraph 2, lines 39-41)

“*Statistical analysis.* The binding data were analyzed using Student t-test, **one-way ANOVA and nonlinear regression fitting to saturation** ($Y = \frac{B_{max} * X}{(K_d + X)}$) **or competition** ($Y = \frac{100}{1 + 10^{((\text{LogEC}_{50} - X) * \text{HillSlope})}}$) **binding curves** with the GraphPad Prism...” (Page 12, paragraph 1, lines 39-44)

References

- Paila, Y. D. & Chattopadhyay, A. The function of G-protein coupled receptors and membrane cholesterol: specific or general interaction? *Glycoconj. J.* **26**, 711–20 (2009).
- Paila, Y. D. & Chattopadhyay, A. Membrane Cholesterol in the Function and Organization of G-Protein Coupled Receptors. *Subcell. Biochem.* **51**, 381–398 (2010).
- Oates, J. & Watts, A. Uncovering the intimate relationship between lipids, cholesterol and GPCR activation. *Subcell Biochem* **21**, 802–7 (2011).
- Gimpl, G. Interaction of G protein coupled receptors and cholesterol. *Chem. Phys. Lipids* **199**, 61–73 (2016).
- Gimpl, G., Burger, K., Politowska, E., Ciarkowski, J. & Fahrenholz, F. Oxytocin receptors and cholesterol: interaction and regulation. *Exp. Physiol.* **85 Spec No**, 41S–49S (2000).
- Prasad, R., Paila, Y. D., Chattopadhyay, A. & Jafurulla, M. Membrane cholesterol depletion enhances ligand binding function of human serotonin1A receptors in neuronal cells. *Biochem. Biophys. Res. Commun.* **390**, 93–6 (2009).
- Pucadyil, T. J. & Chattopadhyay, A. Cholesterol depletion induces dynamic confinement of the G-protein coupled serotonin(1A) receptor in the plasma membrane of living cells. *Biochim. Biophys. Acta* **1768**, 655–68 (2007).
- Prasad, R., Paila, Y. D., Jafurulla, M. & Chattopadhyay, A. Membrane cholesterol depletion from live cells enhances the function of human serotonin(1A) receptors. *Biochem. Biophys. Res. Commun.* **389**, 333–7 (2009).

28. Lagane, B. *et al.* Role of sterols in modulating the human mu-opioid receptor function in *Saccharomyces cerevisiae*. *J. Biol. Chem.* **275**, 33197–200 (2000).
29. Huang, P. *et al.* Cholesterol reduction by methyl- β -cyclodextrin attenuates the delta opioid receptor-mediated signaling in neuronal cells but enhances it in non-neuronal cells. *Biochem. Pharmacol.* **73**, 534–549 (2007).
30. Xu, W. *et al.* Localization of the kappa Opioid Receptor in Lipid Rafts. *J. Pharmacol. Exp. Ther.* **317**, 1295–1306 (2006).
31. Lei, B., Morris, D. P., Smith, M. P. & Schwinn, D. A. Lipid rafts constrain basal alpha-1A-adrenergic receptor signaling by maintaining receptor in an inactive conformation. *Cell. Signal.* **21**, 1532–1539 (2009).
32. Bari, M., Paradisi, A., Pasquariello, N. & Maccarrone, M. Cholesterol-dependent modulation of type 1 cannabinoid receptors in nerve cells. *J. Neurosci. Res.* **81**, 275–283 (2005).
33. Bari, M., Battista, N., Fezza, F., Finazzi-Agrò, A. & Maccarrone, M. Lipid rafts control signaling of type-1 cannabinoid receptors in neuronal cells: Implications for anandamide-induced apoptosis. *J. Biol. Chem.* **280**, 12212–12220 (2005).
34. Colozo, A. T., Park, P. S. H., Sum, C. S., Pisterzi, L. F. & Wells, J. W. Cholesterol as a determinant of cooperativity in the M2 muscarinic cholinergic receptor. *Biochem. Pharmacol.* **74**, 236–255 (2007).
35. Michal, P., Rudajev, V., El-Fakahany, E. E. & Dolezal, V. Membrane cholesterol content influences binding properties of muscarinic M2 receptors and differentially impacts activation of second messenger pathways. *Eur. J. Pharmacol.* **606**, 50–60 (2009).
36. Mitchell, D. C., Straume, M., Miller, J. L. & Litman, B. J. Modulation of Metarhodopsin Formation by Cholesterol-Induced Ordering of Bilayer Lipids? *Biochemistry* **29**, 9143–9149 (1990).
37. Rod, R., Segments, O. & Albert, D. Cholesterol Modulation of Photoreceptor Function in Bovine Retinal Rod Outer Segments. *J. Biol. Chem.* **265**, 20727–20730 (1990).
38. Niu, S. L., Mitchell, D. C. & Litman, B. J. Manipulation of cholesterol levels in rod disk membranes by methyl- β -cyclodextrin: Effects on receptor activation. *J. Biol. Chem.* **277**, 20139–20145 (2002).
39. Kirilovsky, J. & Schramm, M. Delipidation of a beta-adrenergic receptor preparation and reconstitution by specific lipids. *J. Biol. Chem.* **258**, 6841–6849 (1983).
40. Ben-Arie, N., Gileadi, C. & Schramm, M. Interaction of the beta-adrenergic receptor with Gs following delipidation. Specific lipid requirements for Gs activation and GTPase function. *Eur. J. Biochem.* **176**, 649–654 (1988).
41. O'Malley, M. A., Helgeson, M. E., Wagner, N. J. & Robinson, A. S. The morphology and composition of cholesterol-rich micellar nanostructures determine transmembrane protein (GPCR) activity. *Biophys. J.* **100**, L11–3 (2011).
42. Naranjo, A. N., McNeely, P. M., Katsaras, J. & Robinson, A. S. Impact of purification conditions and history on A2A adenosine receptor activity: The role of CHAPS and lipids. *Protein Expr. Purif.* **124**, 62–67 (2016).
43. Bocquet, N. *et al.* Real-time monitoring of binding events on a thermostabilized human A2A receptor embedded in a lipid bilayer by surface plasmon resonance. *Biochim. Biophys. Acta - Biomembr.* (2015). doi:10.1016/j.bbamem.2015.02.014
44. Lam, R. S., Nahirney, D. & Duszyk, M. Cholesterol-dependent regulation of adenosine A(2A) receptor-mediated anion secretion in colon epithelial cells. *Exp. Cell Res.* **315**, 3028–35 (2009).
45. Bari, M. *et al.* Effect of lipid rafts on Cb2 receptor signaling and 2-arachidonoyl-glycerol metabolism in human immune cells. *J. Immunol.* **177**, 4971–80 (2006).
46. Nguyen, D. H. & Taub, D. Cholesterol is essential for macrophage inflammatory protein 1 beta binding and conformational integrity of CC chemokine receptor 5. *Blood* **99**, 4298–4306 (2002).
47. Nguyen, D. H. & Taub, D. D. Inhibition of chemokine receptor function by membrane cholesterol oxidation. *Exp. Cell Res.* **291**, 36–45 (2003).
48. Nguyen, D. H. & Taub, D. CXCR4 function requires membrane cholesterol: implications for HIV infection. *J. Immunol.* **168**, 4121–4126 (2002).
49. Gimpl, G., Burger, K. & Fahrenholz, F. Cholesterol as modulator of receptor function. *Biochemistry* **36**, 10959–74 (1997).
50. Harikumar, K. G. *et al.* Differential effects of modification of membrane cholesterol and sphingolipids on the conformation, function, and trafficking of the G protein-coupled cholecystokinin receptor. *J. Biol. Chem.* **280**, 2176–2185 (2005).
51. Potter, R. M., Harikumar, K. G., Wu, S. V. & Miller, L. J. Differential sensitivity of types 1 and 2 cholecystokinin receptors to membrane cholesterol. *J. Lipid Res.* **53**, 137–48 (2012).

52. Maguire, P. A. & Druse, M. J. The influence of cholesterol on synaptic fluidity, dopamine D1 binding and dopamine-stimulated adenylate cyclase. *Brain Res. Bull.* **23**, 69–74 (1989).
53. Pang, L., Graziano, M. & Wang, S. Membrane cholesterol modulates galanin-GalR2 interaction. *Biochemistry* **38**, 12003–12011 (1999).
54. Eroglu, C., Brugger, B., Wieland, F. & Sinning, I. Glutamate-binding affinity of Drosophila metabotropic glutamate receptor is modulated by association with lipid rafts. *Proc. Natl. Acad. Sci. U. S. A.* **100**, 10219–10224 (2003).
55. Kumari, R., Castillo, C. & Francesconi, A. Agonist-dependent signaling by group I metabotropic glutamate receptors is regulated by association with lipid domains. *J. Biol. Chem.* **288**, 32004–32019 (2013).
56. Furukawa, H. & Haga, T. Expression of functional M2 muscarinic acetylcholine receptor in *Escherichia coli*. *J. Biochem.* **127**, 151–61 (2000).
57. Oates, J. *et al.* The role of cholesterol on the activity and stability of neurotensin receptor 1. *Biochim. Biophys. Acta* **1818**, 2228–33 (2012).
58. Monastyrskaya, K., Hostettler, A., Buergi, S. & Draeger, A. The NK1 receptor localizes to the plasma membrane microdomains, and its activation is dependent on lipid raft integrity. *J. Biol. Chem.* **280**, 7135–7146 (2005).
59. Levitt, E. S., Clark, M. J., Jenkins, P. M., Martens, J. R. & Traynor, J. R. Differential effect of membrane cholesterol removal on mu and delta opioid receptors: A parallel comparison of acute and chronic signaling to adenylyl cyclase. *J. Biol. Chem.* (2009).
60. Gaibelet, G. *et al.* Cholesterol content drives distinct pharmacological behaviours of micro-opioid receptor in different microdomains of the CHO plasma membrane. *Mol. Membr. Biol.* **25**, 423–35 (2008).
61. Gimpl, G., Reitz, J., Brauer, S. & Trossen, C. Oxytocin receptors: ligand binding, signalling and cholesterol dependence. *Prog. Brain Res.* **170**, 193–204 (2008).
62. Fahrenholz, F., Klein, U. & Gimpl, G. Conversion of the myometrial oxytocin receptor from low to high affinity state by cholesterol. *Adv Exp Med Biol* **395**, 311–319 (1995).
63. Klein, U., Gimpl, G. & Fahrenholz, F. Alteration of the myometrial plasma membrane cholesterol content with beta-cyclodextrin modulates the binding affinity of the oxytocin receptor. *Biochemistry* **34**, 13784–13793 (1995).
64. Pucadyil, T. J. & Chattopadhyay, A. Cholesterol modulates the antagonist-binding function of hippocampal serotonin1A receptors. *Biochim. Biophys. Acta* **1714**, 35–42 (2005).
65. Chattopadhyay, A., Jafurulla, M., Kalipatnapu, S., Pucadyil, T. J. & Harikumar, K. G. Role of cholesterol in ligand binding and G-protein coupling of serotonin1A receptors solubilized from bovine hippocampus. *Biochem. Biophys. Res. Commun.* **327**, 1036–41 (2005).
66. Pucadyil, T. J. & Chattopadhyay, A. Cholesterol modulates ligand binding and G-protein coupling to serotonin(1A) receptors from bovine hippocampus. *Biochim. Biophys. Acta* **1663**, 188–200 (2004).
67. Sjögren, B., Hamblin, M. W. & Svenningsson, P. Cholesterol depletion reduces serotonin binding and signaling via human 5-HT7(a) receptors. *Eur. J. Pharmacol.* **552**, 1–10 (2006).
68. Thompson, A. A. *et al.* GPCR stabilization using the bicelle-like architecture of mixed sterol-detergent micelles. *Methods* **55**, 310–317 (2011).
69. Zoicher, M., Zhang, C., Rasmussen, S. G. F., Kobilka, B. K. & Müller, D. J. Cholesterol increases kinetic, energetic, and mechanical stability of the human β 2-adrenergic receptor. *Proc. Natl. Acad. Sci. U. S. A.* **109**, E3463–72 (2012).
70. Gimpl, G. & Fahrenholz, F. Cholesterol as stabilizer of the oxytocin receptor. *Biochim. Biophys. Acta - Biomembr.* **1564**, 384–392 (2002).
71. O'Malley, M. A., Helgeson, M. E., Wagner, N. J. & Robinson, A. S. Toward rational design of protein detergent complexes: Determinants of mixed micelles that are critical for the in vitro stabilization of a g-protein coupled receptor. *Biophys. J.* **101**, 1938–1948 (2011).
72. Albert, A. D., Boesze-Battaglia, K., Paw, Z., Watts, A. & Epan, R. M. Effect of cholesterol on rhodopsin stability in disk membranes. *Biochim. Biophys. Acta - Protein Struct. Mol. Enzymol.* **1297**, 77–82 (1996).
73. Jastrzebska, B., Debinski, A., Filipek, S. & Palczewski, K. Role of membrane integrity on G protein-coupled receptors: Rhodopsin stability and function. *Prog. Lipid Res.* **50**, 267–77 (2011).
74. Howell, S. C. *et al.* CHOBIMALT: A cholesterol-based detergent. *Biochemistry* **49**, 9572–9583 (2010).
75. Kruse, A. C. *et al.* Activation and allosteric modulation of a muscarinic acetylcholine receptor. *Nature* **504**, 101–6 (2013).
76. Mazella, J., Chabry, J., Kitabgi, P. & Vincent, J. P. Solubilization and characterization of active

- neurotensin receptors from mouse brain. *J. Biol. Chem.* **263**, 144–149 (1988).
77. Kulig, W. *et al.* How well does cholesteryl hemisuccinate mimic cholesterol in saturated phospholipid bilayers? *J. Mol. Model.* **20**, (2014).
 78. Kimura, T. *et al.* Recombinant cannabinoid type 2 receptor in liposome model activates G protein in response to anionic lipid constituents. *J. Biol. Chem.* **287**, 4076–4087 (2012).
 79. Sýkora, J., Bouřová, L., Hof, M. & Svoboda, P. The effect of detergents on trimeric G-protein activity in isolated plasma membranes from rat brain cortex: Correlation with studies of DPH and Laurdan fluorescence. *Biochim. Biophys. Acta - Biomembr.* **1788**, 324–332 (2009).
 80. Boesze-Battaglia, K., Hennessey, T. & Albert, A. D. Cholesterol heterogeneity in bovine rod outer segment disk membranes. *J. Biol. Chem.* **264**, 8151–8155 (1989).
 81. Boesze-Battaglia, K., Fliesler, S. J. & Albert, A. D. Relationship of cholesterol content to spatial distribution and age of disk membranes in retinal rod outer segments. *J. Biol. Chem.* **265**, 18867–18870 (1990).
 82. Schultz, Z. D. Raman spectroscopic imaging of cholesterol and docosahexaenoic acid distribution in the retinal rod outer segment. in *Australian Journal of Chemistry* **64**, 611–616 (2011).
 83. Brown, M. F. Modulation of rhodopsin function by properties of the membrane bilayer. *Chem. Phys. Lipids* **73**, 159–180 (1994).
 84. Albert, A. D. & Boesze-Battaglia, K. The role of cholesterol in rod outer segment membranes. *Progress in Lipid Research* **44**, 99–124 (2005).
 85. Albert, A., Alexander, D. & Boesze-Battaglia, K. Cholesterol in the Rod Outer Segment: A Complex Role in a ‘Simple’ System. *Chem. Phys. Lipids* **199**, 94–105 (2016).
 86. Soubias, O. & Gawrisch, K. The role of the lipid matrix for structure and function of the GPCR rhodopsin. *Biochim. Biophys. Acta* **1818**, 234–40 (2012).
 87. Bennett, M. P. & Mitchell, D. C. Regulation of membrane proteins by dietary lipids: effects of cholesterol and docosahexaenoic acid acyl chain-containing phospholipids on rhodopsin stability and function. *Biophys. J.* **95**, 1206–1216 (2008).
 88. Mahammad, S. & Parmryd, I. Cholesterol homeostasis in T cells. Methyl- β -cyclodextrin treatment results in equal loss of cholesterol from Triton X-100 soluble and insoluble fractions. *Biochim. Biophys. Acta - Biomembr.* **1778**, 1251–1258 (2008).
 89. Zidovetzki, R. & Levitan, I. Use of cyclodextrins to manipulate plasma membrane cholesterol content: Evidence, misconceptions and control strategies. *Biochim. Biophys. Acta - Biomembr.* **1768**, 1311–1324 (2007).
 90. Roehm, N. W., Rodgers, G. H., Hatfield, S. M. & Glasebrook, A. L. An improved colorimetric assay for cell proliferation and viability utilizing the tetrazolium salt XTT. *J. Immunol. Methods* **142**, 257–265 (1991).
 91. Barducci, A., Bussi, G. & Parrinello, M. Well-tempered metadynamics: A smoothly converging and tunable free-energy method. *Phys. Rev. Lett.* **100**, (2008).
 92. Wang, T. & Duan, Y. Chromophore channeling in the G-protein coupled receptor rhodopsin. *J. Am. Chem. Soc.* **129**, 6970–1 (2007).
 93. Wang, T. & Duan, Y. Ligand entry and exit pathways in the beta2-adrenergic receptor. *J. Mol. Biol.* **392**, 1102–15 (2009).
 94. Miao, Y., Feher, V. A. & McCammon, J. A. Gaussian Accelerated Molecular Dynamics: Unconstrained Enhanced Sampling and Free Energy Calculation. *J. Chem. Theory Comput.* **11**, 3584–3595 (2015).
 95. Kappel, K., Miao, Y. & McCammon, J. A. Accelerated molecular dynamics simulations of ligand binding to a muscarinic G-protein-coupled receptor. *Q. Rev. Biophys.* **48**, 479–487 (2015).
 96. Saleh, N. *et al.* A Three-Site Mechanism for Agonist/Antagonist Selective Binding to Vasopressin Receptors. *Angew. Chemie* **55**, 8008–8012 (2016).

Reviewer #1 (Remarks to the Author)

The revised version of the manuscript has fully addressed my concerns. I would like to thank the authors for taking time to carry out additional experiments and analyses as requested.

The only point that needs clarification, in my opinion, is related to Figure S18 in the Supplement. Since I cannot find any methodological description in the text (main or supplement) of how in silico model of MTSEA-B attached to the three cysteine residues was performed, I was not sure what the green surface in Figure S18 represents. Is it one of the many possibilities that MTSEA-B can be covalently linked? Or is it some kind of ensemble of structures? I believe it is important to show range of conformations the reagent can take once covalently attached.

Reviewer #2 (Remarks to the Author)

The authors have satisfactorily responded to all my questions and made the necessary changes to the manuscript.

Reviewer #3 (Remarks to the Author)

The authors have clearly put a substantial amount of effort into their response, and overall the effect is quite convincing. I believe that some aspects of their methodology and results deserve to be emphasized, given the growing interest in experimental approaches to cholesterol-dependent GPCR function. In particular, I encourage the authors to highlight the lipidomics measurements of the changes in membrane composition following MBCD treatment, and the fact that the cells remain viable despite depletion of 60% of their membrane cholesterol. Indeed, this last result is the exception rather than the rule, and therefore should be emphasized.

Response to reviewers

Reviewer#1

Remarks to the Author:

The revised version of the manuscript has fully addressed my concerns. I would like to thank the authors for taking time to carry out additional experiments and analyses as requested.

We thank again Reviewer 1 time she/he invested in carefully reviewing the final version of this manuscript as well as for her/his valuable comments.

R1.1) *The only point that needs clarification, in my opinion, is related to Figure S18 in the Supplement. Since I cannot find any methodological description in the text (main or supplement) of how in silico model of MTSEA-B attached to the three cysteine residues was performed, I was not sure what the green surface in Figure S18 represents. Is it one of the many possibilities that MTSEA-B can be covalently linked? Or is it some kind of ensemble of structures? I believe it is important to show range of conformations the reagent can take once covalently attached.*

We completely agree with the reviewer that Figure S18 (now Supplementary Fig. 15) needs further clarifications. In addition, we acknowledge that the methodological description of the modeling task carried out here was missing.

In short, we carried out a low-mode type conformational search using the LowModeMD method¹ in the MOE package (Chemical Computing Group) which uses a specific molecular dynamics protocol to sample low-strain energy conformations. After each simulation run, the current MTSEA-B conformation is extracted, optimized and stored in a database. This method yielded around 100 different conformers for each of the three cysteine substitutions. The green surface we showed in Figure S18 represented the average structure of all conformers obtained using LowModeMD. However, we agree that it is important to show the conformational space that MTSEA-B reagent can adopt once covalently attached to the three cysteine residues. Therefore, we have modified Figure S18 (now Supplementary Fig. 15) to depict all conformers (green stick) obtained during the modeling process. In addition, we added a detailed method description into the main manuscript as follows:

Methods

“In silico model and conformational analysis for MTSEA-B chemical modification. In a first step, the side chain of cysteines (C3.30, C5.46 and C6.56) of the A_{2A}R (PDB:3EML) was chemically modified by attaching MTSEA-B using the builder tool of the MOE package (version 2016.08). In a second step, the conformational space of such chemical modification was explored using the LowModeMD method in the MOE package with: Rejection Limit 100, Iteration Limit 100, RMS Gradient 0.1, MM Iteration Limit 500, RMSD Limit 0.5, Energy Window 100, Conformation Limit 1000 and applying the Amber10:EHT force field. As a result of this search, we obtained 98 different conformers for the chemically modified residues C3.30 and C5.46 and 96 conformers for the chemically modified residue C6.56, as shown in Supplementary Fig. 15.” (Page 11, paragraph 2)

References

1. Labute, P. LowModeMD - Implicit low-mode velocity filtering applied to conformational search of macrocycles and protein loops. *J. Chem. Inf. Model.* **50**, 792–800 (2010).

Reviewer#2

Remarks to the Author:

The authors have satisfactorily responded to all my questions and made the necessary changes to the manuscript.

Thanks again for your careful review.

Reviewer#3

Remarks to the Author:

The authors have clearly put a substantial amount of effort into their response, and overall the effect is quite convincing. I believe that some aspects of their methodology and results deserve to be emphasized, given the growing interest in experimental approaches to cholesterol-dependent GPCR function.

Again, we appreciate and thank the time and critical comments provided by Reviewer 3.

R3.1) *In particular, I encourage the authors to highlight the lipidomics measurements of the changes in membrane composition following MBCD treatment, and the fact that the cells remain viable despite depletion of 60% of their membrane cholesterol. Indeed, this last result is the exception rather than the rule, and therefore should be emphasized.*

According to the suggestion of Reviewer#3, we have further emphasized the lipidomics measurements of the changes in membrane composition and the cell viability measurements following MBCD treatment as following:

Results

~~“As shown in Supplementary Fig. S12~~Remarkably, 40 min treatment with 5 mM M β CD according to the described protocol in the method section depletes up to 61% of cholesterol from the membrane (Supplementary Fig. 1).” (Page 3, paragraph 2, lines 8-10)

“...affected cell viability. Here it is worth to highlight that cells remain viable despite the depletion of more than 60% of their membrane cholesterol using the M β CD treatment detailed in the Methods. Likewise,...” (Page 3, paragraph 3, lines 3-5)